# Transmembrane chemokines act as receptors in a novel mechanism termed inverse signaling

Kirsten Hattermann[1†], Henrike Gebhardt[1†], Sebastian Krossa[2], Andreas Ludwig[3], Ralph Lucius[1], Janka Held-Feindt[4*‡], Rolf Mentlein[1*‡]

[1]Department of Anatomy, University of Kiel, Kiel, Germany; [2]Department of Structural Biology, Institute of Zoology, Kiel, Germany; [3]Institute for Pharmacology and Toxicology, RWTH Aachen University, Aachen, Germany; [4]Department of Neurosurgery, University Medical Center Schleswig-Holstein, Kiel, Germany

**Abstract** The transmembrane chemokines CX3CL1/fractalkine and CXCL16 are widely expressed in different types of tumors, often without an appropriate expression of their classical receptors. We observed that receptor-negative cancer cells could be stimulated by the soluble chemokines. Searching for alternative receptors we detected that all cells expressing or transfected with transmembrane chemokine ligands bound the soluble chemokines with high affinity and responded by phosphorylation of intracellular kinases, enhanced proliferation and anti-apoptosis. This activity requires the intracellular domain and apparently the dimerization of the transmembrane chemokine ligand. Thus, shed soluble chemokines can generate auto- or paracrine signals by binding and activating their transmembrane forms. We term this novel mechanism "inverse signaling". We suppose that inverse signaling is an autocrine feedback and fine-tuning system in the communication between cells that in tumors supports stabilization and proliferation.

*For correspondence: janka.held-feindt@uksh.de (JHF); rment@anat.uni-kiel.de (RM)

†These authors contributed equally to this work
‡These authors also contributed equally to this work

**Competing interests:** The authors declare that no competing interests exist.

## Introduction

The chemokines CX3CL1/fractalkine and CXCL16 are transmembrane (*tm*) proteins (*Bazan et al., 1997*; *Matloubian et al., 2000*) that are converted to soluble ligands by metalloproteinases, in particular ADAM10 and ADAM17 (a disintegrin and metalloproteinase, *Garton et al., 2001*; *Hundhausen et al., 2003*; *Abel et al., 2004*; *Ludwig et al., 2005*). These soluble (*s-*) chemokines induce signal transduction and chemotaxis in target cells which express their classical G protein-coupled receptors CX3CR1 and CXCR6/Bonzo; however, also firm adhesion between *tm*-chemokine- and chemokine-receptor-bearing cells has been reported (*Imai et al., 1997*; *Wilbanks et al., 2001*). Of note, both chemokines themselves, show potential signal transduction sites in their short intracellular domains, but so far signaling via the transmembrane forms has not been described (*Bazan et al., 1997*; *Matloubian et al., 2000*).

In contrast, several transmembrane ligands themselves are known to transduce signals after binding of their respective receptor. This reverse transduction of signals by transmembrane ligands has been initially observed with ligands of the tumor necrosis factor (TNF) family after binding to their cognate receptors (or to antibodies), and thus bidirectional responses are produced (*Ferran et al., 1994*; *Eissner et al., 2004*; *Lettau et al., 2011*). This principle has been also described for IL-15 (*Neely et al., 2004*), ephrin-ligands (*Klein, 2009*) and semaphorins (*Zhou et al., 2008*). Reverse signaling can induce several biological functions, e. g. co-stimulation, silencing, transmission of additional signals (*Eissner et al., 2004*; *Sun and Fink, 2007*), or synapse formation and plasticity in the nervous system (*Klein, 2009*). Reverse signaling may thus be an alternative to an autocrine-signaling

**eLife digest** The cells that make up an animal need to communicate with each other for a variety of purposes, including controlling the growth and repair of tissues. Commonly, such signaling involves 'ligand' molecules binding to specific 'receptor' proteins embedded in the cell membrane. When a ligand docks to the right receptor protein, the parts of the receptor inside the cell change shape. This activates signaling pathways within that cell.

Types of ligands called transmembrane ligands are found embedded in cell membranes. Some cancer cells have high levels of transmembrane ligands called CXCL16 and CX3CL1 but do not produce the corresponding receptors for these molecules. The part of these ligands that sits outside of the cells can also be separated from the rest of the molecule to produce a soluble ligand that can move around outside the cell.

By studying cancer cells using microscopy and biochemical approaches, Hattermann, Gebhardt et al. now show that the soluble forms of CXCL16 and CX3CL1 bind to their transmembrane equivalents. This activates signaling pathways that promote cell growth and make the cancer cells more resistant to cell death. However, this signaling did not occur if the transmembrane ligands were altered to lack the part normally found inside the cell, which suggests that transmembrane CXCL16 and CX3CL1 act as receptors.

It was not previously known that a soluble ligand could activate its transmembrane equivalent. Hattermann, Gebhardt et al. have named this process "inverse signaling", and suggest that it helps to fine-tune the communication between cells. Future experiments will need to study the importance of inverse signaling in living animals and investigate how it works alongside other signaling methods.

loop with classical receptors. So far, reverse signaling has only been reported for distinct transmembrane ligand molecules although signaling motifs for reverse signaling are more abundant and also found within the transmembrane variants of the chemokine family.

Investigating the role of transmembrane chemokines in cancer, we detected a remarkable high synthesis in several types of tumor cells without an appropriate expression of their classical receptors (*Held-Feindt et al., 2010*; *Hattermann et al., 2013*). Therefore, we looked for alternative functions and receptors. We found that cells expressing high levels of a *tm*-chemokine also responded to their soluble counterparts without expressing the respective G protein-coupled chemokine receptor. Further detailed investigations revealed that *tm*-chemokines themselves, in fact, responded to soluble chemokines via a novel mechanism by binding their soluble forms and subsequently inducing signals and functional responses. We propose that this novel mechanism leads to auto- or paracrine activation of cells expressing *tm*-chemokines that either function as surface receptor or can be shed to generate soluble ligands of another *tm*-chemokine on the cell surface.

## Results

### Transmembrane chemokines are highly expressed in several types of tumor cells

By quantitative RT-PCR we detected high levels of CXCL16 and CX3CL1 in human glioma, neuroblastoma, colon carcinoma, lower levels in breast cancer cells, and LOX melanoma cells produced very low or non-detectable mRNA amounts (*Figure 1* upper panel; $\Delta C_T$ in logarithmic scale, thus a 3.3 higher $\Delta C_T$ value indicates a 10-fold lower expression). Beside tumor cells, also endothelial cells and monocytes like THP-1 cells express *tm*-chemokines. In contrast to this broad distribution of ligands, the expression of the receptors CXCR6 and CX3CR1 was restricted to only a few cell types, e.g. activated T-cells (positive control, compare *Ludwig et al., 2005*) or monocytes/monocytic THP-1 cells. The expression of CX3CL1 and CXCL16 was confirmed on protein level by immunocytochemistry (lower panel of *Figure 1*), which revealed high expression for the brain tumor cell lines and low expression for the breast carcinoma cell line MCF-7. The LOX melanoma cell line showed no protein expression, and therefore was chosen as a control cell line in the subsequent experiments. CXCR6 and CX3CR1 were not detectable in the solid tumor cell lines on protein level either (not shown) by

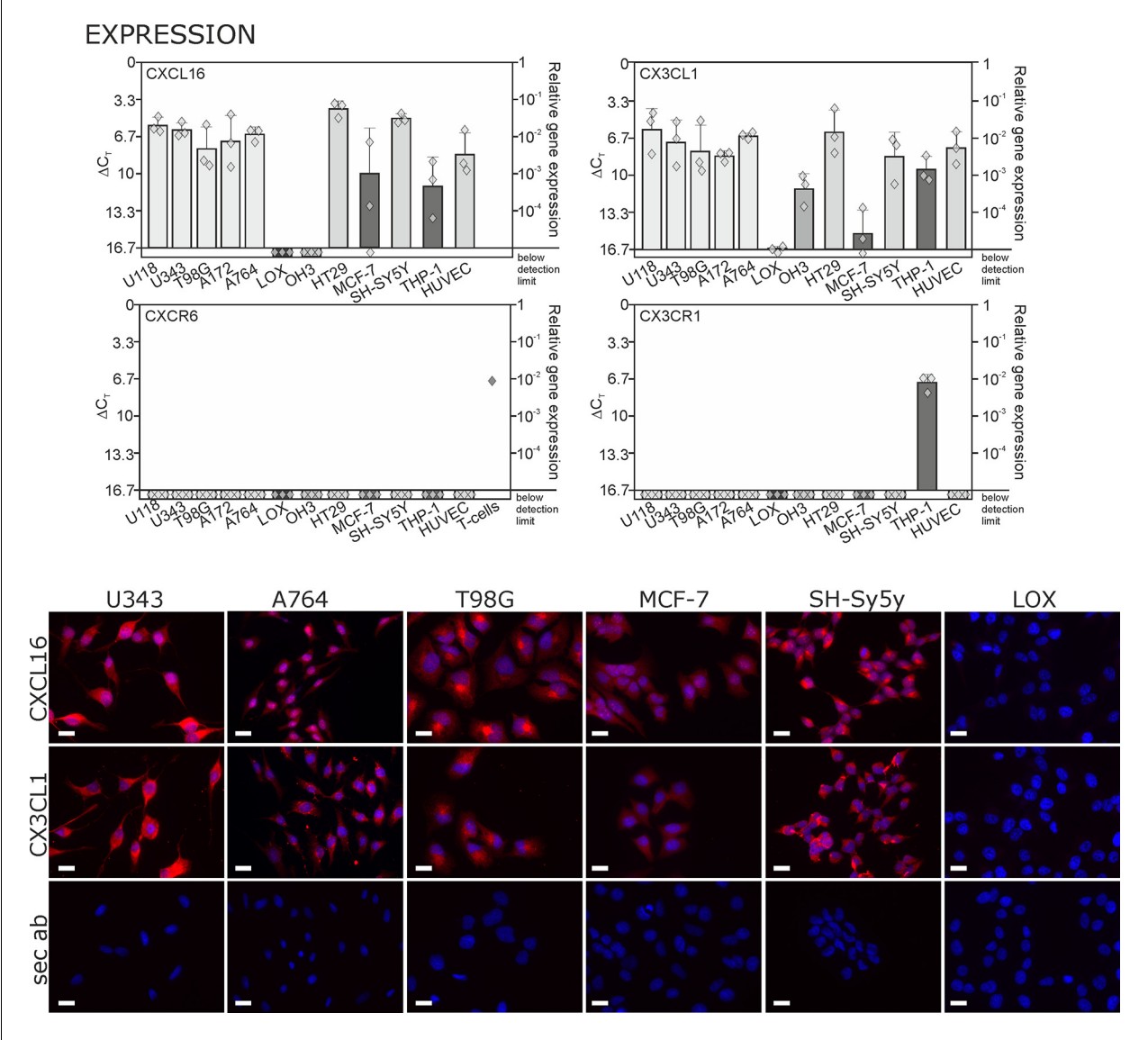

**Figure 1.** Expression of transmembrane chemokines and their known receptors in various cell types. **Top:** As determined by qRT-PCR, the transmembrane chemokines CXCL16 and CX3CL1 are highly transcribed in many human tumor cell lines including glioma (U118, U343, T98G, A172, A764), colon carcinoma (HT29)and neuroblastoma cells (SH-SY5Y), in monocytes (THP-1) and in endothelial cells (HUVEC), at lower levels in breast cancer cells (MCF-7), but not/negligible in LOX melanoma. OH3 small cell lung cancer cells produced CX3CL1, but not CXCL16. In contrast, the known receptors CXCR6 or CX3CR1 were only detectable in a sample of activated T cells or in THP-1 cells, but not in tumor or endothelial cells (n = 3 biological replicates, single data indicated by diamonds). **Bottom:** Immunostaining of a selection of tumor cells exemplarily confirms cell specific protein expression levels of the transmembrane chemokines, and their absence in LOX melanoma cells. Micrographs were taken with exposure times of 600 ms (CXCL16) or 800 ms (CX3CL1, secondary antibody control [sec ab]) for each cell line. Bars indicate 20 µm, n = 3 independent experiments.

immunocytochemistry using antibodies that were suitable for this application in recent investigations (*Held-Feindt et al., 2010*; *Hattermann et al., 2013*). Although these axes of transmembrane chemokines together with their receptors can play important roles in selected tumor-stroma cell interactions, the circumscribed lack of receptors in brain tumor cells themselves contrasts the broad and high expression of chemokines that has also been shown for gliomas and schwannomas *in* situ (*Held-Feindt et al., 2010*; *Hattermann et al., 2013*; *Held-Feindt et al., 2008*).

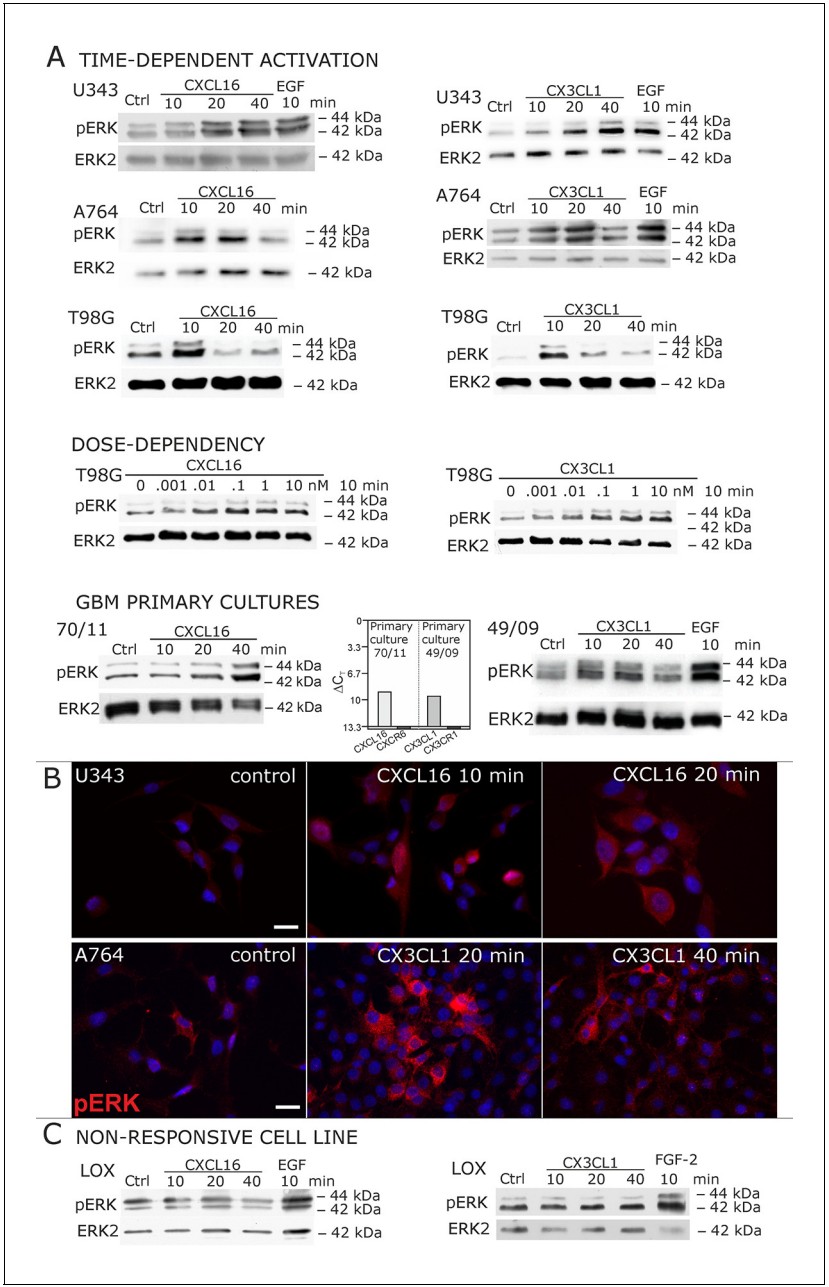

**Figure 2.** Signal transduction in receptor-negative (CXCR6⁻, CX3CR1⁻) tumor cells after stimulation with soluble chemokines (1 nM *s*-CXCL16 or *s*-CX3CL1). A) As shown by Western blots after SDS-PAGE separation, receptor-negative but hence responsive cell lines like the glioma cell lines U343, A764, T98G and primary glioma cultures from different patients display a time- and dose-dependent phosphorylation of the kinase ERK (extracellular signal-regulated kinase p42/p44) after stimulation with *s*-chemokines for the indicated times (compare also *Figure 2 - figure supplement 1*). The responsiveness coincidences with the presence or absence of the corresponding *tm*-chemokines; compare *Figure 1*. Stimulation with epidermal growth factor (EGF; 2 nM) serves as a positive phosphorylation control. Re-blot against non-phosphorylated kinase ERK2 ensures equal loading of the lanes. (B) Immunostaining of *s*-chemokine-stimulated U343 or A764 glioma cells confirms the time-dependent phosphorylation of ERK (rabbit anti-pERK 1/2 with secondary antibody Alexa-Fluor 555 (red)-anti-rabbit IgG; blue nuclear counterstaining with DAPI). Bars indicate 20 µm. (C): The *tm*-chemokine negative LOX melanoma cells are not responsive to 1 nM *s*-chemokines, ERK-phosphorylation is only observed in positive control samples stimulated with epidermal or fibroblast growth factors (EGF or FGF-2, 2 nM). All shown data are representative results from 2-3 independent experiments, respectively, for biological replicates please refer to *Figure 2—figure supplements 1* and *2*.

The following figure supplements are available for figure 2:

**Figure supplement 1.** Biological replicates of western blot experiments showing time- and dose-dependent activation of ERK1/2 upon stimulation with s-chemokines in responsive glioma cell lines (compare *Figure 2A*)

*Figure 2 continued on next page*

*Figure 2 continued*

**Figure supplement 2.** Biological replicates of western blot experiments showing stimulations of non-responsive LOX melanoma cells with s-chemokines (compare *Figure 2C*)

## Receptor-negative, *tm*-chemokine-expressing cells can be stimulated by soluble chemokines

When cells negative for the known receptors CXCR6 and CX3CL1 (compare *Figure 1*) were stimulated with s-CXCL16 or s-CX3CL1, we could, however, detect a transient, dose-and time-dependent phosphorylation of kinases and further biological effects (*Figures 2A,B* and *3*). In all tested cells that expressed *tm*-chemokines (like U343, A172, A764, T98G glioma cell lines, primary glioma cells, or HT29 colon carcinoma cells; not all shown) phosphorylation of ERKs (extracellular signal-regulated kinase, p42/p44) occurred after 10-40 min using concentrations from 0.01 to 10 nM with maximum responses at about 1 nM. By contrast, in *tm*-chemokine-negative and receptor-negative LOX melanoma cells no signal transduction was observed (*Figure 2C*). These signal transduction effects on *tm*-chemokine-positive cells did not depend on the source of recombinant s-chemokines (not shown). Furthermore, not only cell lines but also primary cells isolated from surgically dissected human gliomas were responsive, which express – as the cell lines – *tm*-chemokines, but not their classical receptors (*Figure 2A*).

To ensure that the observed signal transduction also triggers biological effects, we investigated the influence of s-chemokines on cell proliferation, apoptosis and migration (*Figure 3*). Both chemokines, s-CXCL16 and s-CX3CL1, significantly enhanced proliferation of classical receptor-negative, *tm*-chemokine-positive cells like U343 or A764 glioma cells, whereas double-negative cells like LOX melanoma cells did not respond. Furthermore, stimulation with s-CXCL16 and s-CX3CL1 reduced caspase-3/7 activity that was induced by exposure to Temozolomide, a clinically used chemotherapeutic for gliomas. However, the classical biological effect of chemokines, namely cell migration, was not mediated by this signaling (*Figure 3* bottom, and *Figure 3—source data 1*, *Figure 3—figure supplement 1*).

As a first hint to cell specificity and expression dependent response of the measured effects, the breast cancer cell line MCF-7, that displayed low expression levels of *tm*-chemokines (compare *Figure 1*), showed only a slight phosphorylation signals for ERK1/2 and a slightly reduction of caspase 3/7 activity (induced by Staurosporine) upon stimulation with the s-chemokines (*Figure 3—figure supplement 1*).To exclude that signal transduction and biological effects after stimulation with s-chemokines occurred through activation of other chemokine receptors, we inhibited the classical chemokine receptor $G_{i/o}$-signaling in responsive glioma cells using *Pertussis* toxin. Pre-incubation with *Pertussis* toxin did not influence signal transduction of responsive *tm*-chemokine-positive cells (*Figure 4A*). Its inhibitory effect was confirmed in stimulations of CX3CR1[+]-THP-1 cells. Also, an engineered peptide receptor-antagonist of fractalkine (*Hermand et al., 2008*) mediated ERK phosphorylation in primary glioma cells comparable to the recombinant soluble peptide (*Figure 4A*, lower part). We additionally analyzed the expression profile of a broad panel of CXC-chemokine and decoy receptors. As shown in *Figure 4B*, glioma cell lines did not express CXCR3, CXCR4, CXCR6, CX3CR1, but high levels of CXCR7, which is a receptor for CXCL11 and CXCL12 and can mediate G-protein independent signals via arrestin (*Odemis et al., 2012*). However, inhibition of CXCR7 by the synthetic antagonist (*Hattermann et al., 2010*; *Yamaguchi et al., 2014*) CCX733 exhibited no influence on *tm*-chemokine signaling (*Figure 4C*). As chemokine decoy receptors (D6, DARC, CCX-CKR) were expressed in glioma as well as LOX cells, they could be ruled out as mediators for *tm*-chemokine effects that were absent in the melanoma cells. A viral encoded putative receptor for CX3CL1, US28 (*Matlaf et al., 2013*), was detected neither in glioma cells, nor in LOX melanoma cells, and thus, can be excluded for s-CXCL16 or s-CX3CL1 signaling. Additionally, the expression of CD44, a receptor for hyaluronic acid and further ligands, e.g. CCL5 (*Roscic-Mrkic et al., 2005*)was determined at comparable protein levels in glioma cells and melanoma cells. These findings as well as independence of s-chemokine-mediated signal transduction from *Pertussis* toxin suggest that classical G protein-coupled chemokine receptors are not involved in the described effects of s-chemokines. Furthermore, related CXC-receptors and chemokine decoy receptors are absent or can be

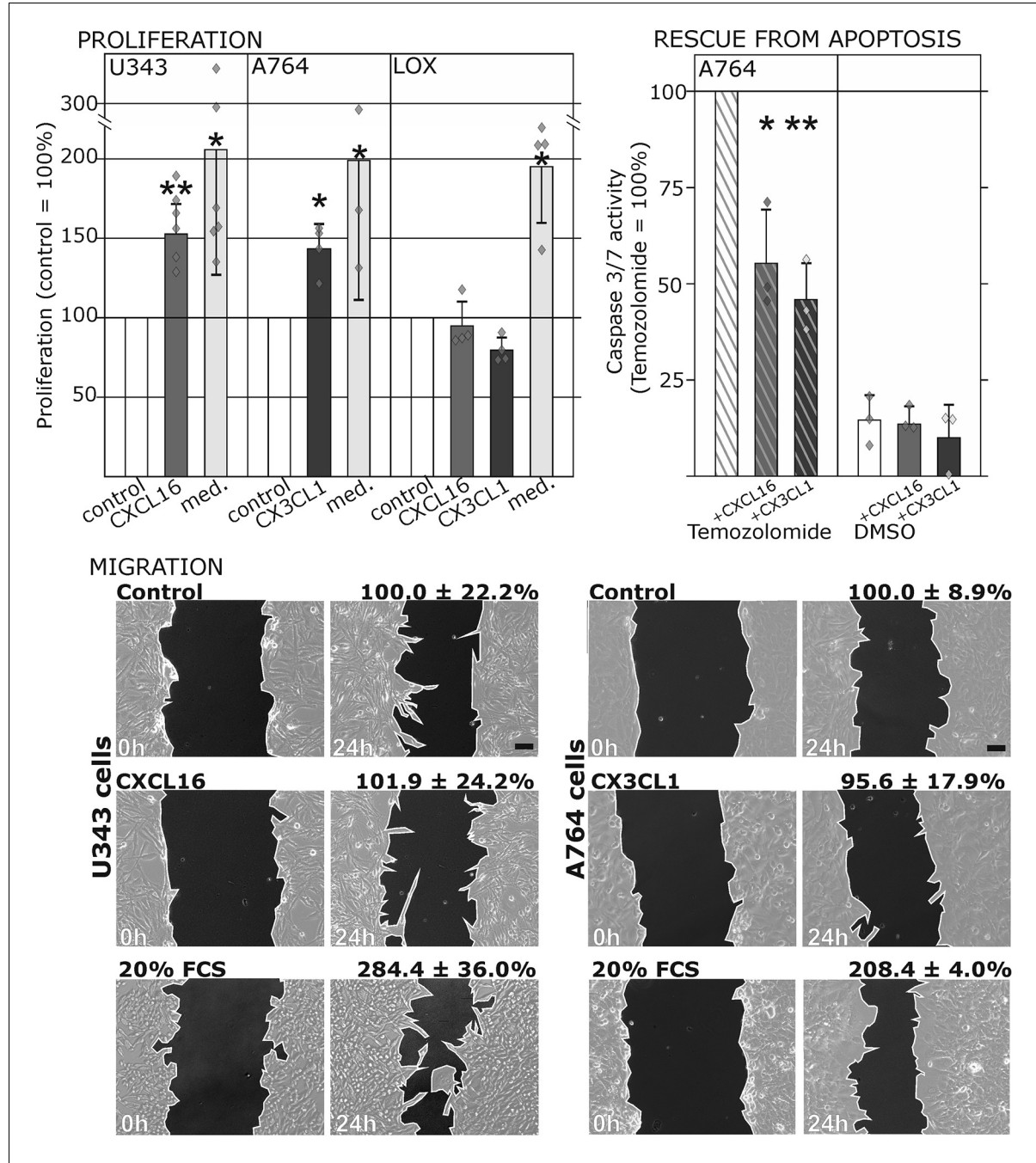

**Figure 3.** Biological effects after stimulation of receptor-negative (CXCR6-, CX3CR1-) tumor cells with soluble chemokines (1 nM s-CXCL16 or s-CX3CL1). **Top:** Soluble chemokines (1 nM) enhance proliferation of U343 and A764 glioma cells expressing the transmembrane counterparts, but not of LOX melanoma cells that are *tm*-chemokine negative. Stimulation was performed for 24 hr and proliferation analyzed by WST-assay. As positive control, growth medium (med.) with 10% fetal calf serum (FCS) was used. Mean values ± standard deviations of at least 3 independent biological replicates (indicated as diamonds) are shown. Moreover, both *s*-chemokines (1 nM, respectively) reduced caspase-3/7 activity evoked by the chemotherapeutic Temozolomide (400 µg/ml, from stock solution in dimethylsulfoxide, DMSO). Stimulations were performed for 48 hr, controls were supplemented with 2% DMSO (corresponding to the solvent concentration in Temozolomide-stimulated samples). Caspase 3/7 activity was measured by fluorescence of the converted substrate in 3 independent biological replicates. Mean values ± standard deviations are shown. For effects of the tm-chemokine low expressing cell line compare *Figure 3—figure supplement 1*. **Bottom:** Migration of U343 and A764 glioma cells was not influenced by stimulation with 10 nM *s*-CXCL16 or *s*-CX3CL1 in a wound healing ('scratch') assay performed for the indicated times. 20% FCS served as positive control. Mean values ± standard deviations are shown from 2-3 independent biological replicates, for data of biological replicates compare *Figure 3—source data 1*. Representative images are shown, bars indicate 50 µm.

*Figure 3 continued on next page*

*Figure 3 continued*

The following source data and figure supplement are available for figure 3:

**Source data 1.** Biological replicates of the scratch assay shown in *Figure 3*.
**Figure supplement 1.** Only slight activation and effects upon *s*-chemokine stimulation of *tm*-chemokine low expressing MCF-7 cells.

excluded as functional receptors for *s*-CXCL16 or *s*-CX3CL1 as well as other types of receptors that require $G_{i/o}$-association.

These experiments show that soluble chemokines (1) elicit signal transduction and biological effects in cells that do not express their known receptors, (2) the effects are independent from *Pertussis* toxin-sensitive G-proteins and other known chemokine receptors including different decoy receptors, (3) are observed only in cells which express *tm*-chemokines, and (4) occur within a (patho) physiologically relevant concentration range and time frame. Therefore, responsive cells must have either novel receptors, or even likely respond to *s*-chemokines via their *tm*-chemokines.

## Transmembrane chemokines bind soluble chemokines

To identify receptors for *s*-chemokines, we performed binding experiments with labeled peptides (*Figure 5A,B and C*). Fluorescent- or biotin-labeled *s*-CXCL16 or *s*-CX3CL1 bound to cells that express their *tm*-chemokine counterparts (as shown by fluorescence microscopy *Figure 5A* and FACS analysis *Figure 5B*), but not to *tm*-chemokine negative LOX cells. However, when LOX cells were transfected with *tm*-CXCL16 or *tm*-CX3CL1, binding of *s*-CXCL16 or *s*-CX3CL1 was clearly observed. Fluorescent or biotin-labeled control peptides (e.g. lactalbumin conjugates) did not bind to these cells (not shown).

The direct association of *s*- to *tm*-chemokines was also visualized by immuno-electron microscopy (*Figure 5C*). For this, *tm*-chemokines were natively stained in glioma cells, *tm*-chemokine transfected and non-transfected LOX melanoma cells with respective antibodies and secondary antibodies labeled with 15 nm gold particles. Subsequently, cells were incubated either with *s*-CXCL16 directly coupled with 5 nm colloidal gold or with biotinylated *s*-CX3CL1 followed by a conjugate of streptavidin and 5 nm gold particles. An association of the different-sized types of colloidal gold particles (exemplarily shown in *Figure 5C*) clearly visualizes binding of the *s*-chemokine to cell-anchored *tm*-chemokine.

A direct association of *s*-chemokines to *tm*-chemokines could also be verified by chemical cross-linking (*Figure 5D*). Isolated membranes from *tm*-chemokine transfected LOX cells were incubated with their corresponding *s*-chemokine (omitted in controls) and cross-linked by exposure to paraformaldehyde (PFA); then proteins were separated by SDS-PAGE and blots incubated with antibodies to soluble chemokines. Only after cross-linking in the presence of *s*-chemokines a shift of the bands of *tm*-chemokines to higher molecular masses was observed, corresponding roughly to the binding of 1-2 soluble peptides. It should be noted that also soluble chemokines alone polymerize under these conditions with cross-linker (not shown).

These experimental approaches show that *tm*-chemokines bind to their soluble counterparts at nanomolar concentrations and *tm*-chemokines and *s*-chemokines appear in close proximity on the cell surface.

## Transmembrane chemokines transduce signals - proof of principle: overexpression and silencing of *tm*-chemokines

To further verify our hypothesis we transfected non-responsive *tm*-chemokine negative LOX cells with expression vectors for *tm*-CXCL16 or *tm*-CX3CL1 and investigated activation of *tm*-chemokine mediated signal transduction (*Figure 6*). Upon stimulation with soluble chemokines, transiently (not shown) as well as stably transfected cells showed a time dependent phosphorylation of ERK 1/2 (*Figure 6A*), which was not the case in non-transfected cells (compare *Figure 2*).

As the short intracellular domain of *tm*-chemokines could be involved in this signal transduction, we performed corresponding experiments with C-terminally truncated *tm*-chemokines that lacked the complete intracellular but not the transmembrane domain (*Figure 6B*). In fact, this truncation abolished the signal transduction effects observed with the full-length *tm*-chemokines.

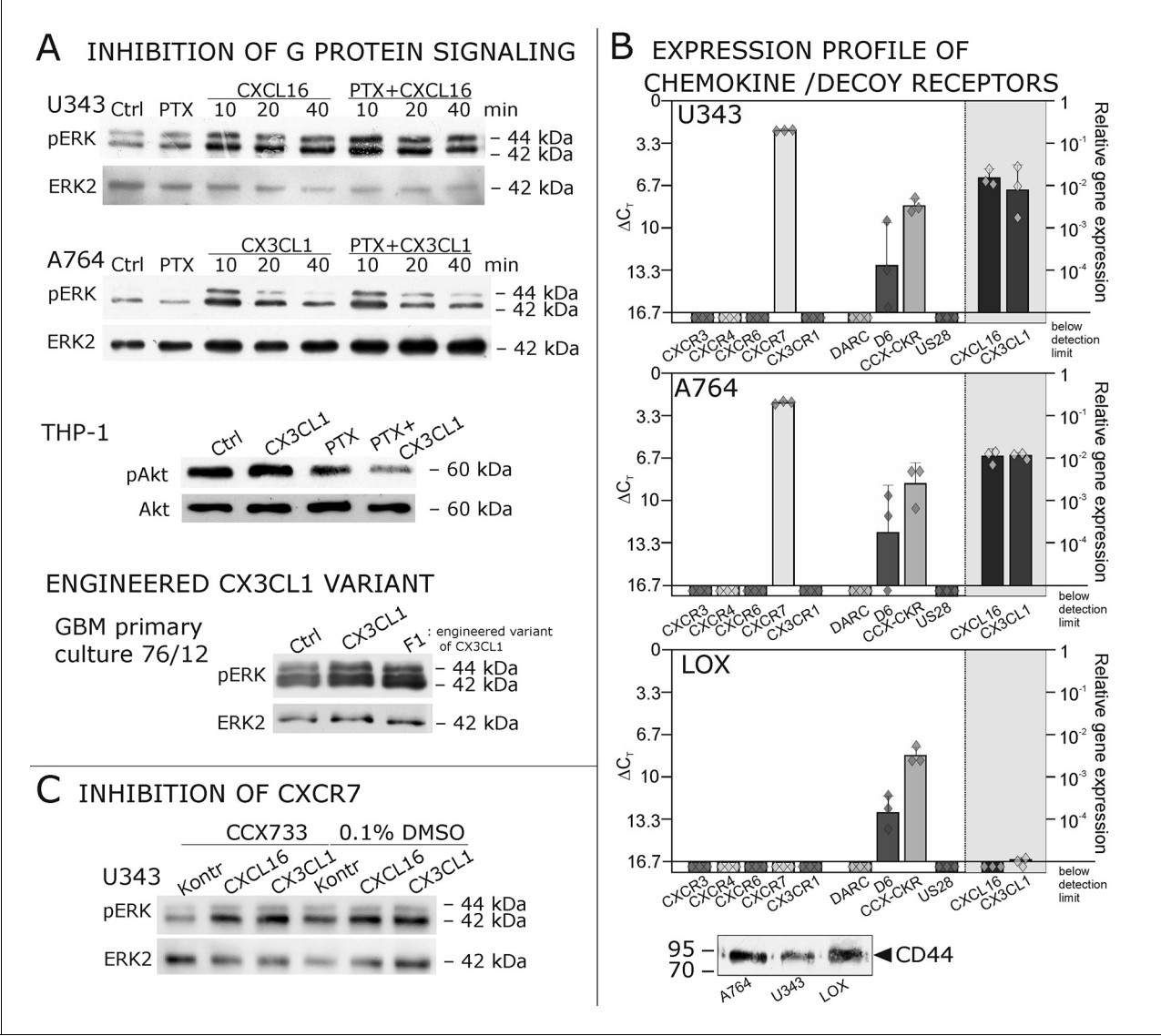

**Figure 4.** Inhibition experiments and transcription analysis exclude the involvement of other chemokine receptors. (**A**) *Pertussis* toxin (PTX, 200 ng/ml) inhibiting $G_{i/o}$-signaling of classical chemokine receptors has no effect on *s*-chemokine-mediated phosphorylation of kinases in U343 or A764 cells. However, in CX3CR1-expressing THP-1 cells (compare *Figure 1*) CX3CL1 mediated phosphorylation of Akt (stimulation with 1 nM for 20 min) can be inhibited by pre-incubation with PTX. An engineered variant of CX3CL1, the recombinant CX3CR1-antagonist F1 (100 nM, 20 min) induces also signal transduction in primary glioma cells indicating a mechanism different from CX3CR1-binding (and antagonism). Shown are representative Western blots after SDS-PAGE separation of lysates from stimulated U343 or A764 glioma cells stained for pERK 1/2 or pAkt (re-blot to non-phosphorylated kinases, control of equal loading) from 3 independent experiments, compare *Figure 4—figure supplement 1*. (**B**) The transcription profile of classical chemokine receptors and chemokine decoy receptors as determined by quantitative RT-PCR shows that the chemokine receptors CXCR3, CXCR4, CXCR6 and CX3CR1 are absent in responsive U343 and A764 and non-responsive LOX cells. However, the atypical chemokine receptor CXCR7 that is known to signal G protein-independently is expressed in responsive cell lines and absent in LOX cells. The chemokine decoy receptors D6 and CCX-CKR are expressed at comparable levels in responsive and non-responsive cells, whereas DARC is absent (n = 3 biological replicates, indicated by diamonds). Additionally, the cytomegalovirus-derived gene US28 that encodes for a putative CX3CR1 receptor could not be detected in these cell lines. The highly glycosylated protein CD44, that may putatively sequester chemokines, was detected at comparable protein levels in U343, A764 and LOX cells (n = 3). (**C**) To investigate the contribution of CXCR7 in *s*-chemokine mediated signaling U343 cells were pre-incubated with the CXCR7-antagonist CCX733 (100 nM, 2 hr) and stimulated with 1 nM of *s*-CXCL16 or *s*-CX3CL1 for 20 min. Controls ± *s*-chemokines were pre-incubated with 0.1% DMSO as solvent controls. The CXCR7-antagonist CCX733 does not impair *s*-chemokine-mediated ERK phosphorylation. Representative Western blots from 3 independent experiments. For biological replicates of western blots please refer to *Figure 4—figure supplement 1*.

The following figure supplement is available for figure 4:

**Figure supplement 1.** Biological replicates of western blot experiments with s-chemokines and Pertussis toxin (compare *Figure 4A*).

Next, to exclude experimental artifacts from recombinant *s*-chemokines, we wanted to verify if the naturally occurring *s*-chemokines, which are released by proteolytic cleavage from their *tm*-forms, could also elicit *tm*-chemokine-signaling. Since transfected cells should release *s*-chemokines by shedding over time, we tested first their occurrence in conditioned medium, and then assayed if conditioned, *s*-chemokine-containing media could induce signals in overexpressing cells that were washed prior to experiments. In fact, 24 hr- or 48 hr-conditioned media of *tm*-chemokine-transfected cells, but not of vector-transfected cells, contained considerable amounts of corresponding *s*-chemokines as detected by Western blot or ELISA (*Figure 6C*). As expected, conditioned media from transfected, but not from control cells, induced ERK 1/2 phosphorylation; again non- or control-transfected LOX cells yielded no signal transduction (*Figure 6D*).

Regarding biological effects of *s*-chemokines in *tm*-chemokine-transfected LOX cells we could observe a rescue from Camptothecin-induced cell death events. As shown by immunocytochemistry and Western blot, stimulation of *tm*-CXCL16-transfected LOX cells with *s*-CXCL16 and *tm*-CX3CL1-transfected LOX cells with *s*-CX3CL1 significantly reduced the amount of the cleaved Poly (ADP-Ribose) Polymerase (PARP), whereas mock-transfected LOX cells were not rescued (*Figure 7*).

As contrary approach to overexpression in non-responsive, *tm*-chemokine negative cells, silencing of *tm*-chemokines in responsible cells can further prove our concept. Therefore, we reduced *tm*-chemokines in responsive receptor-negative glioma cell lines and primary cultures from surgical samples by siRNA and stimulated with the corresponding soluble chemokines (*Figure 8*). By siRNA silencing, *tm*-chemokine transcript amounts were reduced up to 24-59% of controls transfected with unspecific control siRNAs. The *s*-chemokine-mediated ERK-phosphorylation could be observed in control siRNA-treated glioma cell lines and primary cultures but was clearly reduced in *tm*-chemokine silenced cells.

Antibodies specific for the ligand portion are known to activate signaling of transmembrane ligands such as of *tm*-TNF-α(*Eissner et al., 2004*probably as they can mediate the required di/multimerization. To find out, whether antibodies against the chemokine domains of CXCL16 or CX3CL1 can also induce signaling of the *tm*-chemokines, we stimulated stably transfected *tm*-CXCL16 or *tm*-CX3CL1 LOX cells with 0.1 µg/ml of the respective antibody (*Figure 9A*); IgG from the same species and mock-transfected cells served as controls. In fact, specific antibodies, but not controls induced ERK1/2 phosphorylation in *tm*-chemokine expressing cells. To test if the activation potential depends on di-/oligomerization of the *tm*-chemokines, we compared the efficiency of complete to monovalent antibodies. However, monovalent F(ab) fragments of the same antibodies generated by papain digestion and purification did not produce any effects in transfected cells (*Figure 9B*). These experiments show that antibodies like soluble ligands induce signal transduction through *tm*-chemokines, and this process seems to depend on di/oligomerization of the *tm*-chemokines.

In summary, transfection as well as silencing experiments evidence that transmembrane chemokines transduce signals when stimulated with nanomolar concentrations of corresponding soluble counterparts exerting also biological effects as exemplarily shown by rescue from apoptosis. Thereby, signaling is elicited by recombinant peptides (chemokine domain) or soluble chemokines produced by transfected cells themselves. Furthermore, the intracellular C-terminal domains of *tm*-chemokines are required for these effects. Furthermore, specific chemokine domain-directed antibodies, but not monovalent fragments induce signaling emphasizing the necessity of dimerization of the *tm*-chemokines to induce signaling. Thus, *tm*-chemokines do not only bind *s*-chemokines, but also induce a signal transduction and further biological effects specifically.

## Discussion

Reverse signaling of transmembrane ligands together with classical receptor signaling is an important mode of bidirectional signaling, both in the nervous and immune systems. We here provide evidence for a novel form of auto- and paracrine signaling of transmembrane ligands produced by their shed, soluble ligands – a process that we term 'inverse signaling'. Whereas reverse signaling requires the binding of the transmembrane ligand to its classical membrane-bound or soluble receptor, the inverse signaling depends on binding of the soluble ligand to its transmembrane counterpart.

This binding of the soluble forms of the chemokines CXCL16 and CX3CL1 to their transmembrane counterparts - as a prerequisite for inverse signaling - was verified by FACS analysis, morphological assays at the light- and electron-microscopic levels and chemical cross-linking experiments. Similarly,

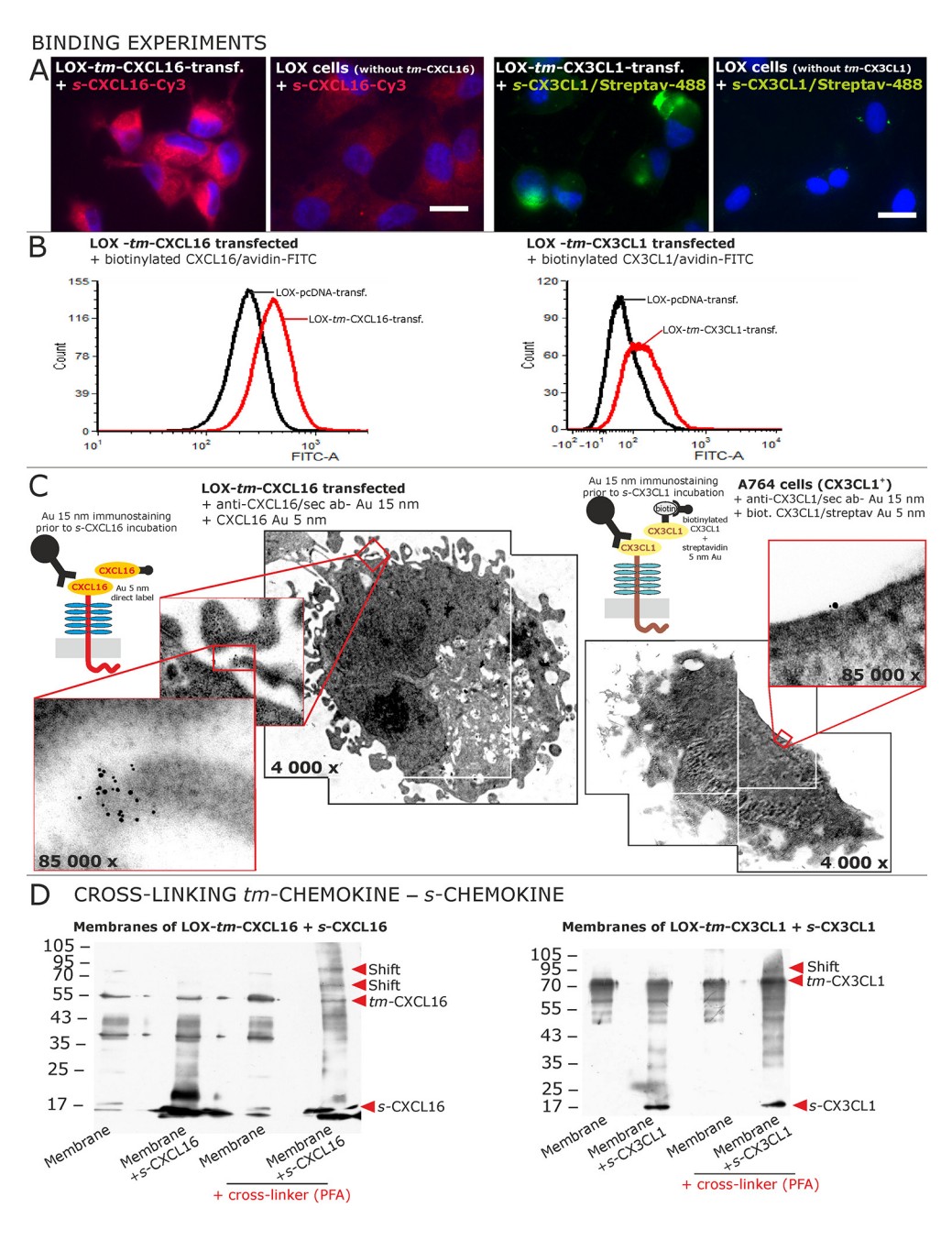

**Figure 5.** Binding of soluble chemokines to corresponding *tm*-chemokines on cell surfaces. (**A**) Fluorescent soluble chemokines, either labeled directly (cyanine3, Cy3) or indirectly via biotin-label to fluorescent (strept)avidin (Alexa-Fluor 488 or fluorescein-isothiocyanate, FITC), bind to tumor cells expressing *tm*-chemokines (transfected LOX melanoma, or glioma cells, not shown), but not to non-transfected LOX cells as shown by light microscopy. Due to its lipophilic character, the dye Cy3 yields a whiff of background. However, this was also observed with the negative control Cy3-labeled lactalbumin. Chemokine concentrations 2 nM, bars indicate 20 μm. (**B**) FACS analysis confirmed binding signals yielded by fluorescence microscopy. LOX cells transfected with *tm*-CXCL16 were detectable by their binding of biotinylated CXCL16 (2 nM)/Avidin-FITC, whereas mock transfected cells (LOX-pcDNA) were not, and LOX cells transfected with *tm*-CX3CL1 were labeled by biotin-CX3CL1 (2 nM)/Avidin-FITC, in comparison to mock transfected cells. (**C**) A close association between *s*- and *tm*-chemokines is visible by immuno-electron microscopy of *tm*-chemokine-expressing/-transfected cells which were immunolabeled with anti-CXCL16 or CX3CL1 and 15 nm-gold-labeled secondary antibodies and subsequently incubated with 5 nm-gold-labeled (directly or via biotin/streptavidin complex) soluble chemokines (2 nM) before embedding. (**D**) Chemical cross-linking by paraformaldehyde (PFA) shifts *tm*-chemokine bands to higher molecular weights. Isolated membranes of *tm*-chemokine overexpressing LOX cells were incubated over night with 2 nM *s*-chemokines, cross-linked with 1% PFA and subjected to SDS-PAGE separation and subsequent Western blotting

*Figure 5 continued on next page*

*Figure 5 continued*

using antibodies against the respective chemokine domains of CXCL16 and CX3CL1. All experiments were repeated in 2 independent biological replicates, and representative photographs and immunoblots are shown.

di- and multimerization of the chemokine domain of CX3CL1 had already earlier been shown by X-ray crystal structure (*Hoover et al., 2000*), and is also known for some other (soluble) chemokines.

Next to the ability to bind to *tm*-chemokines, *s*-chemokines may also elicit intracellular signals. This could be clearly shown in primary glioblastoma cells and corresponding cell lines. After transfection with *tm*-chemokine expression vectors, previously non-responsive cells may also exert inverse signaling upon stimulation with *s*-chemokines. Opposite to overexpression, silencing of *tm*-chemokines reduces signaling induced by *s*-chemokine stimulation. Consequently, by two different approaches (overexpression and silencing) signal transduction of the *s*-chemokines was shown to depend on the expression level of transmembrane CXCL16 or CX3CL1 in the responsive cell line. Apart from recombinant peptides, also the naturally shed soluble ligands as contained in conditioned media were able to induce inverse signaling and, as a sign of specificity, antibodies raised against the chemokine domain of CXCL16 or CX3CL1 yielded comparable signals. This activation of *tm*-chemokine signaling by intact antibodies but not by corresponding monovalent F(ab) fragments supports a putative necessity of di/oligomerization of the *tm*-chemokines for the inverse signaling. And indeed, a clustering/bundle formation of transmembrane CX3CL1 has been observed in transfected and primary cells by bioluminescence resonance energy transfer (BRET) and homogeneous time-resolved fluorescence (HTRF; *Hermand et al., 2008*; *Ostuni et al., 2014*).

However, it is not unlikely, that inverse signaling may require a co-receptor supporting the *tm*-chemokines. Some other classical chemokine receptors evaluated are either absent, or can be excluded. For example, inverse signaling is insensitive to *Pertussis* toxin, an inhibitor of classical chemokine receptor signaling via $G_{i/o}$-proteins, and is not affected by inhibition of CXCR7, a non-canonical chemokine receptor signaling via arrestin. However, putative co-receptors (and also intracellular binding partners) need further investigation.

Signaling domains of the intracellular tails of transmembrane ligands seem to be critical for the signal transduction in reverse signaling, and thus also may transduce inverse signaling. For example, TNF-α, FasL and other members of the TNF family, contain S/TXXS/T sequences and proline-rich domains (FasL) that can bind adaptor proteins and thereby transduce signals (*Kennelly and Krebs, 1991*; *Watts et al., 1999*; *Eissner et al., 2004*; *Sun and Fink, 2007*; *Amanchy et al., 2011*; *Daar, 2012*). In contrast, ephrins and semaphorins signal through PDZ-binding motifs and also proline-rich domains (*Klein, 2009*; *Zhou et al., 2008*; *Daar, 2012*). As shown by transfection/stimulation experiments with C-terminally-truncated *tm*-chemokines, the intact intracellular domains are also essential for the inverse signaling. Identification of the exact binding sites for kinases/adaptor proteins are ongoing and beyond the scope of this investigation. Potential activation sites include cytoplasmic SXXS-sequences (comparable to those of the TNF family) and additionally SH2-binding sites (YXPV/R). In CX3CL1, both sites are highly conserved from monotremes through marsupials to highly developed mammals; in the case of CXCL16, they are found – as far as known - in primates and only same mammals (*Table 1*). Thus, the transfer to a rodent *in vivo* model has to be carefully designed. Of note, the reverse signaling of TNF-α has long been described (*Ferran et al., 1994*; *Lettau et al., 2011*; *Eissner et al., 2004*; *Shao and Schwarz, 2011*), but exact mechanisms of further downstream signaling are not yet known. Apparently, there may be an analogy of transmembrane ligand signaling between ligands of the TNF family and transmembrane chemokines that might be elucidated in future investigations.

Concerning the biological consequences of 'non-classical' signaling, reverse signaling in the case of TNF members mediates co-stimulation, direct stimulation, desensitization and migration yielding a fine-tuning in adaptive immunity and a regulatory feedback in innate immunity (*Eissner et al., 2004*; *Sun and Fink, 2007*). Reverse signaling of ephrins triggers cell adhesion or differentiation, in particular in the nervous system, spine and synapse formation, but also in bone modeling (*Klein, 2009*; *Matsuo and Otaki, 2012*; *Yu et al., 2010*), whereas reverse signaling of semaphorins similarly regulates cell guiding and repulsion, especially in the nervous system (*Yu et al., 2010*). As far as we know, inverse signaling of transmembrane chemokines appears to induce mainly autocrine

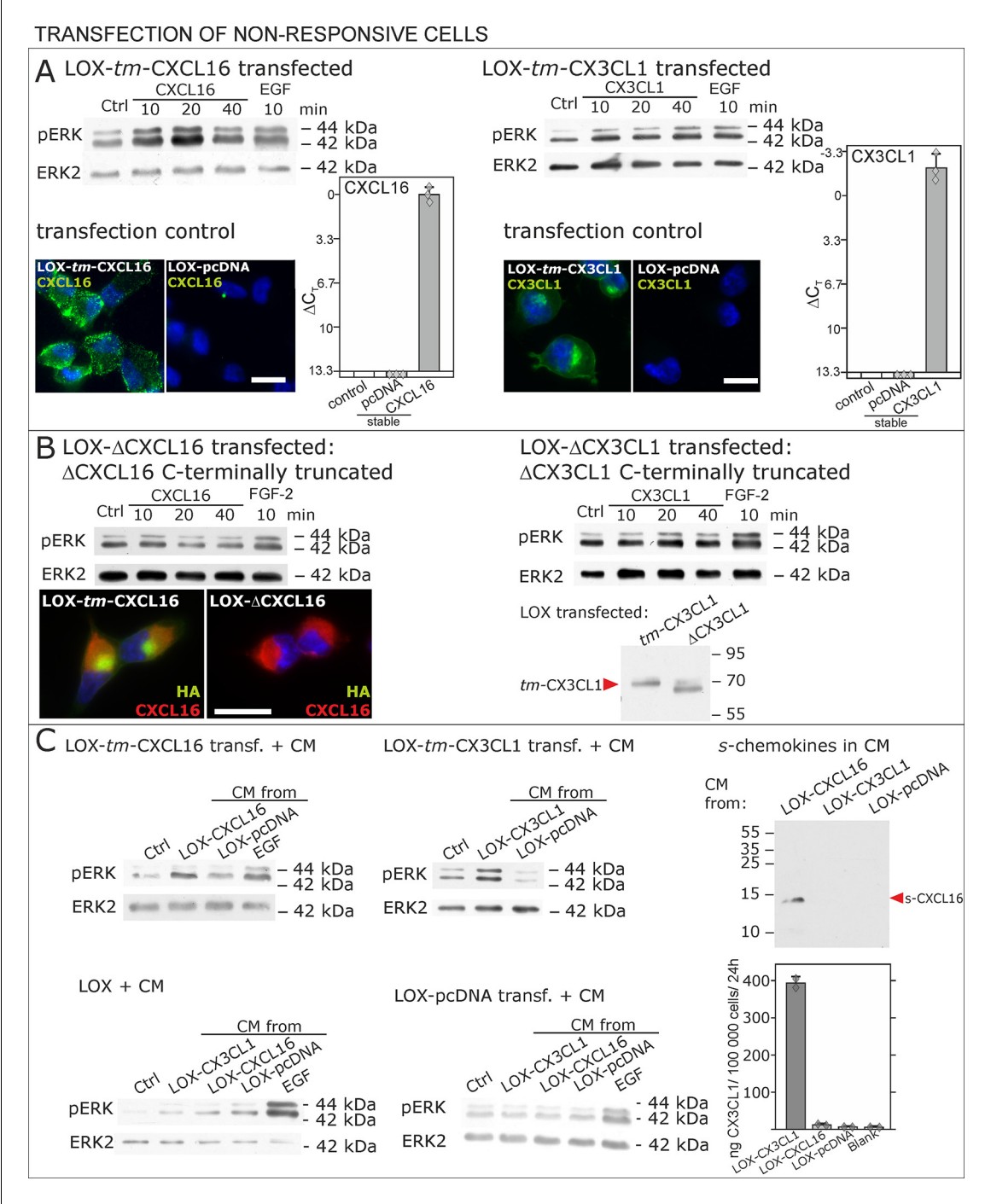

**Figure 6.** Non-responsive cells can be transformed to be responsive to *s*-chemokine stimulation by transfection with *tm*-chemokines. (**A**) LOX melanoma cells were stably transfected with *tm*-chemokines. Transfection efficiency was controlled by quantitative RT-PCR and immunocytochemistry (n = 3 biological replicates as indicated by diamonds; bars represent 20 µm). Cells were then stimulated with *s*-chemokines (1 nM) and cell lysates analyzed by SDS-PAGE separation and immunoblotting for phosphorylation of ERK 1/2 (re-blots for ERK2 ensure equal loading). Transfected cells responded with ERK 1/2-phosphorylation in contrast to non-transfected cells (compare *Figure 2*). (**B**) Stably transfected LOX-cells expressing C-terminally truncated *tm*-chemokine variants lacking the intracellular domain (LOX-ΔCXCL16 and LOX-ΔCX3CL1) cannot be activated by stimulation with *s*-chemokines (1 nM). Thus, the intracellular domain of the *tm*-chemokines seems to be critical for signaling. Successful truncation was proven by co-immunostaining with anti-CXCL16 (extracellular chemokine domain, red) and anti-HA (intracellular tag of the CXCL16-expression vector, green), or band shift in Western blot. To obtain defined bands, proteins were deglycosylated prior to SDS-PAGE. (**C**) Shed chemokine domains found in conditioned media (CM) from overexpressing cells can also mediate activation of *tm*-chemokine transfected LOX cells. Conditioned media were obtained from confluent CXCL16 or CX3CL1 or mock (pcDNA) transfected cells (LOX-CXCL16, LOX-CX3CL1, LOX-pcDNA) and applied to *tm*-chemokine expressing

*Figure 6 continued on next page*

*Figure 6 continued*

LOX cells for 20 min. SDS-PAGE plus immunoblotting (CXCL16) or ELISA (CX3CL1) proved presence of shed chemokines in the conditioned media used for stimulation. Stimulation with conditioned media containing shed chemokine domains can activate the ERK signaling in *tm*-chemokine transfected cells, but not in pcDNA-transfected or not modified LOX cells. All shown results are representatives from 3 independent experiments, except for the control of successful truncation of *tm*-chemokines in stable clones and experiments with conditioned media that were performed twice; for examples of biological replicates compare *Figure 6—figure supplement 1*.

The following figure supplement is available for figure 6:

**Figure supplement 1.** Biological replicates of western blot experiments with transfected LOX melanoma cells (compare *Figure 6*).

stimulatory and stabilizing effects like increased proliferation and anti-apoptosis. These tumor cell protective effects could also be confirmed in transfection experiments enabling a direct comparison of the chemokine effects in *tm*-chemokine expressing versus not-expressing cells of the same cell line. To our knowledge, these effects seem to depend on the *tm*-chemokine expression level rather than the (tumor) cell type. Obviously, tumors would clearly profit from such positive feedback. The importance of autocrine loops in many types of tumors becomes obvious regarding for example growth factors like epidermal (EGF) or platelet-derived (PDGF) growth factors. However, also in inflammation or under non-pathological conditions in tissue development these stabilizing autocrine loops may play important roles.

The expression of transmembrane chemokines can be induced by pro-inflammatory cytokines like TNF-α, IFN-γ and interleukin-1β, e.g. in fibroblasts and endothelial cells (*Garcia et al., 2000*; *Abel et al., 2004*; *Isozaki et al., 2011*) regulating their role in inflammatory (trans-) migration processes. The release of the chemokine domain is mediated by the cell surface proteases ADAM10 and ADAM17 in a constitutive (mainly ADAM10) or cytokine-induced manner (*Abel et al., 2004*; *Ludwig et al., 2005*). ADAM17 was initially discovered as TNF-α-converting enzyme (TACE) liberating soluble cytokine from the transmembrane form. Later, other ADAMs were identified as similar

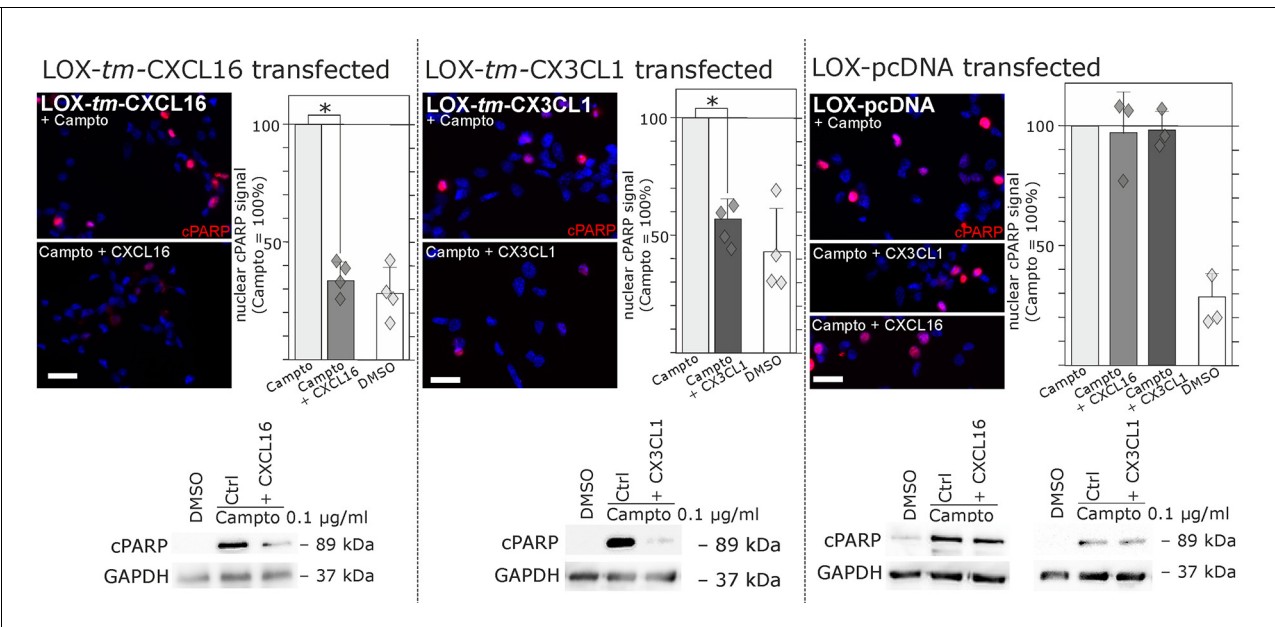

**Figure 7.** Stimulation with *s*-chemokines can mediate rescue from chemically-induced cell death in *tm*-chemokine-transfected LOX melanoma cells. LOX melanoma cells stably expressing *tm*-CXCL16 or *tm*-CX3CL1 (or mock transfected LOX cells) were treated for 18 hr with 0.1 µg/ml Camptothecin (inhibitor of topoisomerase I) to induce cell death. Simultaneous stimulation of *tm*-CXCL16-LOX with 1 nM *s*-CXCL16 or *tm*-CX3CL1-LOX with 1 nM *s*-CX3CL1 significantly reduced cell death as indicated by reduced cleavage of poly(ADP ribose) polymerase (PARP) (shown by Western blot after SDS-PAGE, n = 2 biological replicates; or immunocytochemistry, n = 3-4 biological replicates, indicated by diamonds). In contrast, mock-transfected (pcDNA) LOX cells did not show reduced signals of cleaved PARP when stimulated with *s*-CXCL16 or *s*-CX3CL1. Bars represent 50 µm.

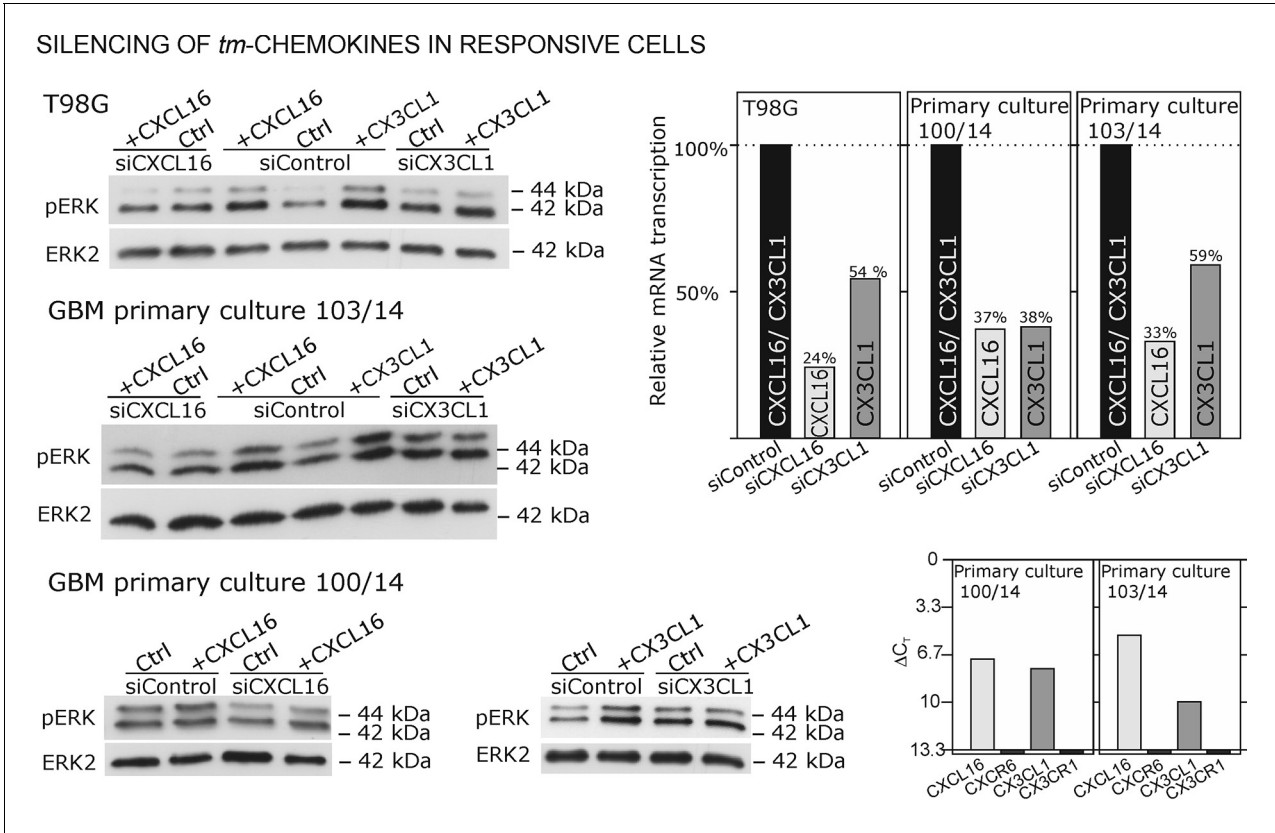

**Figure 8.** Silencing of *tm*-chemokine expression abolishes the *s*-chemokine mediated activation in responsive glioma cell lines and primary cultures. Cell lines and primary cells were transfected with CXCL16 or CX3CL1 specific RNAi (or non-specific control RNAi), left for recovery for 36 hr and stimulated with 1nM CXCL16 or CX3CL1 for 15 min. Samples were analyzed by SDS-PAGE separation and immunoblotting for phosphorylated ERK 1/2, and re-probed for ERK 2 to ensure equal loading. Silencing efficiency and basic transcription level were confirmed by quantitative RT-PCR (right panel). In *tm*-chemokine silenced cultures, no activation can be observed by incubation with the respective *s*-chemokine. Experiments with T98G were repeated in 2 independent biological replicates, shown primary cultures are representative examples from 2 different patient's primary cultures (compare *Figure 8—figure supplement 1*).

The following figure supplement is available for figure 8:

**Figure supplement 1.** Biological replicates of western blot experiments from siRNA knockdown in T98G glioma cells (compare *Figure 8*).

'sheddases', but they also cleave receptors, adhesion molecules and intracellular signaling molecules (*Pruessmeyer and Ludwig, 2009*). ADAMs are supposed to terminate reverse signaling by cleaving transmembrane ligands of the TNF family bound to their receptors (*Sun and Fink, 2007*). In case of ephrin-A, ADAM10 is constitutively associated with the EphA receptor that clusters and is activated upon ligand binding; this positions the proteinase domain for effective ephrin-A5 cleavage and the cleavage occurs in *trans*, with ADAM10 and its substrate being on the membranes of opposing cells (*Janes et al., 2005*). This mechanism ensures that only Eph-bound ephrins are recognized and cleaved. A corresponding regulatory role of ADAMs as for reverse can also be supposed for inverse signaling. Since different physiological and pathophysiological mediators or therapeutics tightly regulate ADAM activities (*Reiss and Saftig, 2009*), inverse signaling should also be modulated in diverse ways in health and disease. However, this question may only be satisfactory addressed in carefully designed future *in vivo* experiments.

A potential regulation of the *tm*-chemokine signaling may also be expected by the interplay of 'inverse' and 'classical' signaling, since both, classical receptors and transmembrane chemokines compete for the same soluble ligand. For example in an inflammatory context, binding of *s*-chemokines to *tm*-counterparts might reduce the recruitment of immune cells expressing the classical receptors, and vice versa might a massive infiltration of *s*-chemokine binding immune cells reduce

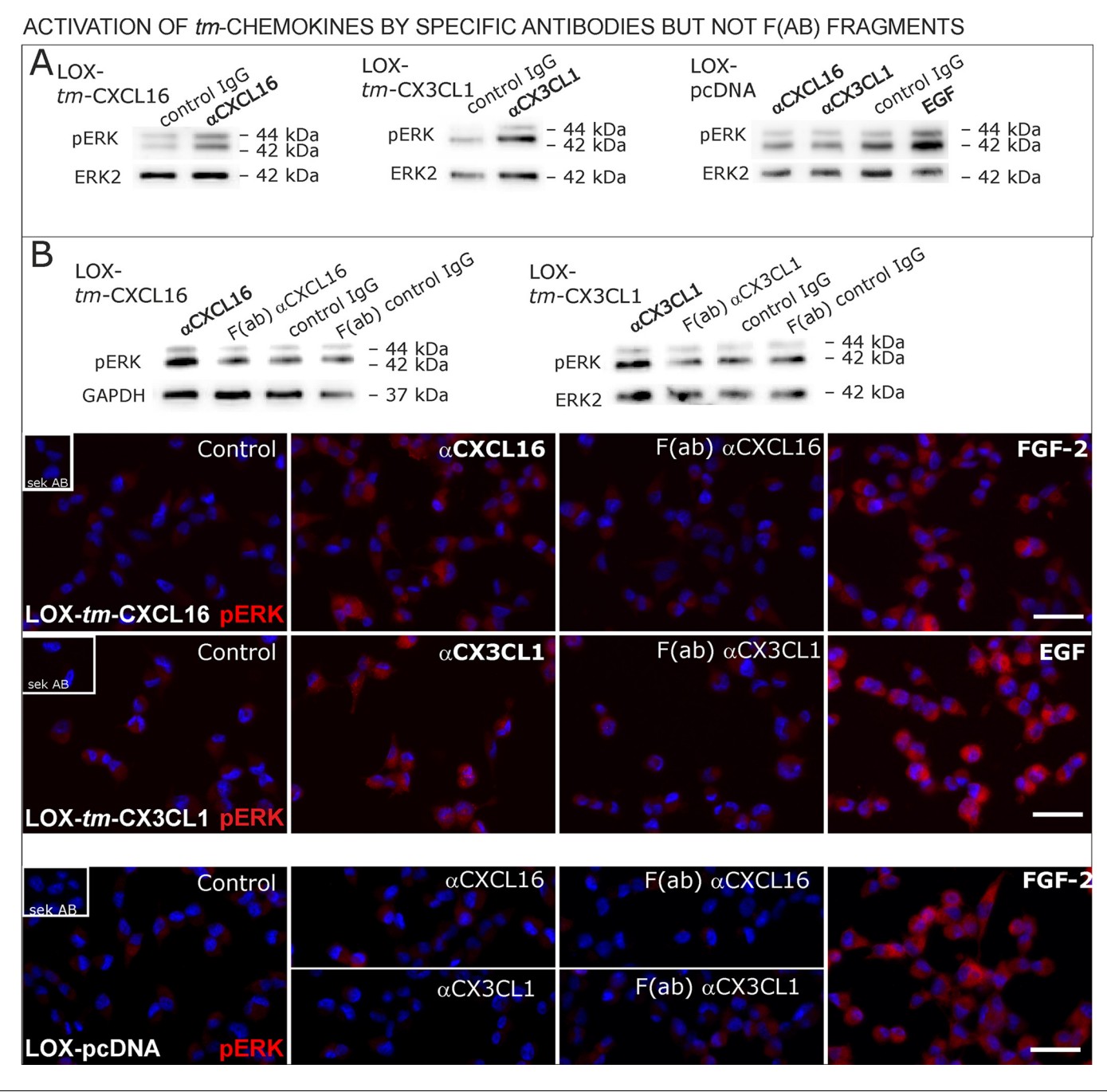

**Figure 9.** Signal transduction in *tm*-chemokine expressing LOX cells upon stimulation with specific antibodies (0.1 µg/ml), but not monovalent F(ab) fragments. (**A**) As shown by Western blot after SDS-PAGE, stimulation of *tm*-CXCL16 or *tm*-CX3CL1 transfected LOX cells for 20 min with antibodies against the corresponding chemokine domains (0.1 µg/ml), yields a phosphorylation signal for ERK1/2. Non-specific control IgG could not elicit signaling, nor could specific antibodies activate mock-transfected (LOX-pcDNA) cells (positive stimulation control: 2 nM FGF-2, n = 2-3 biological replicates, for corresponding effects in glioma cells compare *Figure 9—figure supplement 1*). (**B**) In contrast to the intact specific antibodies, monovalent F(ab) fragments (0.1 µg/ml) obtained by papain digestion and clean-up of these fragments failed to mediate ERK1/2 phosphorylation as demonstrated by Western blot and immunocytochemistry (FGF-2 serves as positive control, n = 3 biological replicates, Bars represent 50 µm).

The following figure supplement is available for figure 9:

**Figure supplement 1.** Biological replicates of western blot experiments of stimulations with chemokine-specific antibodies (compare *Figure 9*).

**Table 1.** Sequences of putative intracellular domains from transmembrane chemokines.

| | CX3CL1 |
|---|---|
| *Homo sapiens* (Human) | -QSLQGCPRKMAGEMAEGLR*YIPR*SCGSNSYVLVPV |
| *Nomascus leucogenys* (Northern white-cheeked gibbon) | -QSLQGCPRKMAGEMAEGLR*YIPR*SCGSNSYVLVPV |
| *Macaca mulatta* (Rhesus macaque) | -QSLQGCP RKMAGEMVEGLR*YIPR*SCGS NSYVLVPV |
| *Bos taurus* (Bovine) | -QRLQSCPHKMVGDVVEGIC*YVPR*SCGSNSYVLVPV |
| *Canis familiaris* (Dog) | -YQSLQGCSR KMAGDMVEGLR*YVPR*SCGSN SYVLVPV |
| *Oryctolagus cuniculus* (Rabbit) | -Q SLQGCPRKMAGEMVEGLR*YV PR*SCGANSYVLVPV |
| *Cavia porcellus* (Guinea pig) | -QSLQGCPRK MAGEMVEGLR*YVPR*SCGSNSYVLVPV |
| *Rattus norvegicus* (Rat) | -QS LQGCPRKMAG EMVEGLR*YVP R*SCGSNSYVLVPV |
| *Mus musculus* (Mouse) | -QSLQGCPRKM AGEMVEGLR*YVPR*SCGSNSYVLVPV |
| *Monodelphis domestica* (Gray short-tailed opossum) | -QSLQSCPRRMAGEVVEGLR*YIPR*SCGSNSYVLVPV |
| *Sarcophilus harrisii* (Tasman devil) | -QSLQSCPRRMAGEVVEGLR*YIPR*SCGSNSYVLVPV |
| *Ornithorhynchus anatinus* (Duckbill platypus) | -QSLQSCPRRMAGEVVEGLR*YIPR*SCGSNSYVLVPV |
| | CXCL16 |
| *Homo sapiens* (Human) | -CKRRRGQSPQSSPD PVH*YIPV*AP DSNT |
| *Gorilla gorilla gorilla* (Lowland gorilla) | -CKRRRGQSPQSSPGLPVH*YIPV*APDSNT |
| *Nomascus leucogenys* (Northern white-cheeked gibbon) | -CKR RRGQSPQSSPDLQFH*YIPV*A PDSNT |
| *Macaca mulatta* (Rhesus macaque) | -CKRRGQSPQSSPDLQLH*YIPV*ASDSNT |
| *Sus scrofa* (Pig) | -CKKRQEQSRQYPPDPQLH*YVPV*ASNINT |
| *Loxodonta africana* (African elephant) | -CKRRREQSRLYYPDLQFH*YKPV*A PDS |
| *Bos taurus* (Bovine) | -C KRRKNQLLQHPPDLAASLYT CSRRTRAENGTL |
| *Equus caballus* (Horse) | -CKKREKTLRPSPDLQAHYERVAPD |
| *Canis familiaris* (Dog) | -CKRREQSLQHPPDLQLH*YTPV*A*SDS*NV |
| *Oryctolagus cuniculus* (Rabbit) | -CKRRRGRSPKYSSGKP |
| *Rattus norvegicus* (Rat) | -CNRRVTRQ*SSSG*LQLC*YTPV* EPRPQGL |
| *Mus musculus* (Mouse) | -CNRRATQQNSAGLQLW*YTPV*EPRP |
| *Myotis lucifugus* (Little brown bat) | -CKRRSKQSPQYSPDLQLQCIPVASYSNS |
| *Ornithorhynchus anatinus* (Duckbill platypus) | -CRRRGAPRNEMLYPQRPKGTSITVQANSPT |

Data from: Uniprot (http://www.uniprot.org/)
http://www.uniprot.org/uniprot/?query=CX3CL1&sort=score
http://www.uniprot.org/uniprot/?query=CXCL16&sort=score
Putative intracellular domains are depicted by homology to the published putative human sequences.
-SXXS- motifs, -SXXT- motifs, -SXS- motifs; -SXPV/R- SH2-binding site

the effects of inverse signaling. Moreover, it can be hypothesized that inverse signaling interacts with signaling of classical receptors, e.g. by adapter proteins. Thus, the observed novel mechanism adds to the understanding of the complex chemokine system.

## Conclusions

Inverse signaling is an autocrine feedback and fine-tuning systems in the communication between cells. Here, soluble ligands that are shed from their transmembrane counterparts induce signals through binding to the transmembrane ligands. Though this was shown here for transmembrane chemokines for the first time, a broader distribution, e.g. in signaling of other transmembrane ligands, and further biological effects have to be further evaluated under normal and pathological conditions *in vitro* and in suitable *in vivo* models.

## Materials and methods

### Peptides and inhibitors

Recombinant human chemokines and growth factors were from PeproTech (Hamburg, Germany), R&D-Systems (Wiesbaden, Germany), or Immunotools (Friesoythe, Germany), *Pertussis* toxin (inhibits G protein-signaling) was from Calbiochem (Merck, Darmstadt, Germany) or Sigma-Aldrich (Munich, Germany). The CX3CR1-antagonist F1, an engineered N-terminally modified recombinant CX3CL1 analogue that binds to CX3CR1 but does not induce signaling, was a kind gift from Prof. Dr. Philippe Deterre, Laboratoire Immunité et Infection, INSERM, Faculté de Médcine Pitié-Salpêtrière, Paris, France (*Hermand et al., 2008*). The synthetic CXCR7-antagonist CCX733 was a kind gift from Dr. Mark E.T. Penfold and Prof. Dr. Thomas J. Schall (ChemoCentryx Inc, Mountain View, CA).

### Cell cultures

Human glioma cell lines A172 (*Giard et al., 1973*; ATCC® CRL-1620; ECACC No. 88062428) and T98G (*Stein, 1979*; ATCC CRL-1690; ECACC No. 92090213) glioma cells were purchased from LGC Standards GmbH (Wesel, Germany), U343-MG (*Westermark et al., 1973*) and U118-MG (*Ponten and Macintyre, 1968*; ATCC HTB-15; identical with the glioma cell line U138; U118 was only used for expression data) were obtained from "Deutsches Krebsforschungszentrum" ("Tumor-bank"; Heidelberg, Germany). Primary glioma cells and the cell line A764 were generated by dissociation from a solid tumor and cultivation (cell line: repeated subcultivation) in Dulbecco's modified Eagle's medium (DMEM; PAN Biotech, Aidenbach, Germany) plus 10% fetal calf serum, FCS. Glioma samples were obtained in accordance with the Helsinki Declaration of 1975 with approval of the ethics committee of the University of Kiel, Germany (file reference: D 442/11 and D 427/15) and after written consent of donors. The mamma carcinoma cell line MCF-7 (*Soule et al., 1973*; ATCC HTB-22; ECACC No. 86012803) cells and the monocytic cancer cell line THP-1 (*Tsuchiya et al., 1982*; ATCC TIM-202; ECACC No. 88081201) were obtained from Cell Line Service (Eppelheim, Germany). The LOX melanoma cell line (established by *Fodstad et al., 1988*; and cited by *Thies et al., 2007*), the HT29 colon carcinoma cell line (described by *Fogh and Trempe, 1957*; and obtained from European Cell Culture Collection, Porton Down, Salisbury, UK; ATCC HTB-38; ECACC No. 91072201), the SH-SY5Y neuroblastoma cell line (established by *Biedler et al., 1978*; and provided by Prof. Dr. Hildebrandt, Institute of Cellular Chemistry, Medical School Hannover, Germany; ATCC CRL-2266; ECACC No. 94030304), and the OH3 small cell lung cancer cell line (established by *Griffin and Baylin, 1985* and cited by *Schumacher et al., 1996*) were gifts from Prof. Dr. Udo Schumacher, Dept. of Anatomy, University of Hamburg, Germany. These cell lines were cultivated in RPMI plus 10% FCS. Preparation of T cells has been described previously (*Ludwig et al., 2005*). HUVEC cells were obtained by Promocell (Heidelberg, Germany), cultivated in Endothelial growth medium (Promocell) and used up to passage 5. All cell lines were kept in a master stock – working stock routine to exclude cross-contaminations and monthly checked for mycoplasma contamination by DAPI staining or PCR (Venor GeM Mycoplasma Detection Kit, Minerva Biolabs, Berlin, Germany). All cell lines were free of mycoplasma contamination.

### Stimulation of cells

For signaling experiments, cells were seeded on 25 cm$^2$ culture flasks, cultivated in medium overnight, washed serum- and chemokine-free (2x, with 1 hr intervals) and stimulated in DMEM plus 0.5% fatty acid-free bovine serum albumin, BSA (Sigma-Aldrich), or RPMI plus 5–10% FCS. Chemokines/growth factors, antibodies or inhibitors were added from stocks in phosphate buffered saline (PBS), or in dimethylsulfoxide (DMSO); in this cases controls were run with a corresponding DMSO concentration (maximum 0.1% ). For *Pertussis* toxin experiments, cells were pre-incubated with the inhibitor overnight, and the concentration (200 ng/ml) maintained during the stimulations. For inhibition of the chemokine receptor CXCR7, the CXCR7-specific antagonist CCX733 was added 2 hr prior to the stimulation and the concentration (100 nM) maintained during the chemokine stimulation. To prepare monovalent F(ab) fragments, 5 µg antibody specific for CXCL16 or CX3CL1 (PeproTech) or control IgG (DAKO, Glostrup, Denmark) were incubated with 800 µg immobilized papain (agarose resins, Pierce/Life technologies, Darmstadt, Germany) for 6 hr at 37°C under gentle shaking (350 rpm). After centrifugation for 3 min at 5000 ×g, 20 µl of protein-A agarose (Santa Cruz, Santa Cruz,

CA) were added to the supernatant and incubated for 1.5 hr at 4°C and shaking (350 rpm). After another centrifugation step (1000 ×g, 5 min), the supernatant was cleaned-up by spinning through a G25-Sephadex column (800 ×g, 2 min; Amersham/GE Healthcare). Cells were stimulated with 0.1 µg/ml F(ab) fragment for 20 min.

## Plasmids, mutations and transfection experiments

Expression vectors for CXCL16-HA (with a hemagglutinin HA tag) and CX3CL1 were established as previously described (*Abel et al., 2004*) in a pcDNA3.1 backbone (Invitrogen, Karlsruhe, Germany), and pcDNA 3.1 was used for control transfections. Transfection was performed with TurboFect (Fermentas, St. Leon-Rot, Germany) in serum-free DMEM without antibiotics using 4 µg of the respective expression vectors with pcDNA3.1 backbone and 4 µl TurboFect in a total volume of 1 ml. After 6 hr, cells were rinsed and normal growth medium (RPMI + 10% FCS) was added. Successful transfection was controlled by immunocytochemistry and/or qRT-PCR. Stable clones were generated by selection with 0.75 mg/ml G418 (Calbiochem), and colonies were picked after 10–20 days, amplified and checked for expression by quantitative RT-PCR and immunocytochemistry. Site directed mutagenesis was performed with the QuikChange Site-Directed Mutagenesis Kit following the manufacturer´s advice (Agilent Technologies, Böblingen, Germany). The complementary primer pairs were designed to generate a double stop codon following the transmembrane domain (MWG-Biotech, AG Ebersberg, Germany; CXCL16: 5'-ctttcctatgtgctg**tgatag**aggaggaggggggcag-3', CX3CL1: 5'-tggccatgttcacctac**tagtaa**ctccagggctgccctcg-3') yielding expression of C-terminally truncated *tm*-chemokines, respectively. Successful mutations were verified by sequencing of the plasmids (GATC Biotech AG, Koblenz, Germany). Surface expression of overexpressed native or truncated *tm*-chemokines was verified by immunocytochemistry. Successful C-terminal truncation of CXCL16 was proven by immunocytochemistry (see below) using antibodies against the extracellular chemokine of CXCL16 (PeproTech, #500-P200) and the HA-tag of CXCL16 (Cell signaling, #2367, clone 6E2). Successful C-terminal truncation of CX3CL1 was proven by band shift in western blot (see below) with an antibody against the extracellular domain of CX3CL1 (Pepro Tech, #500-P98). Therefore, cell membranes were isolated[17] and - to obtain defined bands - subjected to N-deglycosylation by PNGase F (New England Biolabs, Ipswich, MA) following the manufacturer's instructions. Western blot was performed as described below.

## Quantitative RT-PCR

RNA was isolated with the TRIZOL reagent, digested by DNase, cDNA synthesized and real time RT-PCR was performed (*Ludwig et al., 2005*) using TaqMan primer probes (Applied Biosystems, Foster City, CA, USA): *hGAPDH* (Hs99999905_m1), *hCXCL16 (Hs00222859_m1)*, *hCXCR6 (Hs00174843_m1)*, *hCX3CL1 (Hs00171086_m1)*, *hCX3CR1 (Hs00365842_m1)*, *hCXCR3 (Hs00171041_m1)*, *hCXCR4 (Hs00607978_s1)*, *hCXCR7 (Hs00664172_s1)*, *hD6 (Hs01907876_s1)*, *hDARC (Hs01011079_s1)*, *hCCX-CKR (Hs00664347_s1)*. A Custom TaqMan primer probe set was used for the detection of the cytomegalovirus gene encoding protein US28: US28F-5'- CGGCAACT-TCTTGGTGATCTTC-3', US28R-5'- CATCGCCGGAGCATTGA-3', FAM- CCATCACCTGGCGACGT-CGGA-MGB (*Matlaf et al., 2013*). Cycles of threshold ($C_T$) were determined with an ABI PRISM 7000 Sequence detection system. $\Delta C_T$ values = $C_T$Gene of interest - $C_T$GAPDH (glyceraldehyde-3-phosphate dehydrogenase, housekeeping gene). A $\Delta C_T$ value of 3.33 corresponds to one magnitude lower gene expression compared to GAPDH. Biologically independent replicates of cell lines and stable clones were obtained from three independent cultures, e.g. from different passages. Diamonds in the respective graphs indicate biological replicates. Expression data shown for primary cultures and silencing experiments are directly matched to the corresponding stimulation experiments.

## Western blotting

Western blotting was performed as described (*Hattermann et al., 2010*). Briefly, cell lysates (5 µg per lane) were separated by electrophoresis using 10% acrylamide gels, transferred to polyvinylidene fluoride (PVDF) membranes by blotting and incubated with primary antibodies (rabbit anti-phospho-ERK1/2, 1:500, and rabbit anti- phospho-Akt, 1:500, Cell Signaling Technology, Danvers, MA, # 9101 and #406, and mouse-anti-CD44 MEM-85, 1:400, Abcam, Cambridge, UK, ab2212) and afterwards horseradish-peroxidase labeled secondary antibody (goat anti-rabbit, 1:30,000; or goat anti-

mouse, 1:30,000, Santa Cruz, Santa Cruz, CA) followed by chemo-luminescence detection (GE Healthcare, Munich, Germany or Millipore, Darmstadt, Germany). To ensure equal loading amounts membranes were reactivated with methanol, stripped with ReBlot Plus Strong Antibody Strip Solution (Millipore) and re-probed with antibodies against the non-phosphorylated proteins (rabbit anti-ERK2, 1:250; Millipore #05-157 or rabbit anti-Akt, 1:500, Cell Signaling).

## Conditioned media

To obtain conditioned media for stimulation experiments LOX melanoma cells overexpressing CXCL16 or CX3CL1 (or mock transfected control LOX cells) were washed once with PBS and were incubated for 24 hr (CX3CL1)/48 hr (CXCL16) with 3.5 ml DMEM containing 0.5% BSA. The supernatants were centrifuged (5 min, 1500 rpm) to remove cell debris. Soluble CX3CL1 was quantified by ELISA following the manufacturer´s advice (R&D Systems). For CXCL16-quantification, 75 µl of the conditioned media were mixed with 25 µl of SDS-sample-buffer (100 mg/ml SDS, 0.25 M dithiothreitol, 50% glycerin, 0.3 M Tris/HCl pH 6.8 + 0.3% SDS), incubated at 97°C for 10 min and 20 µl were loaded on a 15% acrylamide gel and separated by electrophoresis. Western blotting was performed as described above (rabbit anti-CXCL16, 1:500; PeproTech). Matched experiments of s-chemokine determinations and stimulations were performed two times independently; representatives are shown.

## Immunocytochemistry, binding experiments and electron microscopy

For light microscopy, cells grown on poly-D-lysine-coated cover slips were fixed with ice-cold acetone/methanol (1:1), incubated with antibodies and nuclei counterstained with 4´,6-diamino-2-phenylindole (DAPI) as described (Hattermann et al., 2010). Primary antibodies were: anti-CXCL16 and anti-CX3CL1 (both from rabbit, diluted 1:100 in PBS, PeproTech), and anti-HA (mouse, 1:100; Cell signaling); secondary antibodies: donkey anti-rabbit or donkey anti-mouse IgG conjugated with Alexa Fluor 488 or Alexa 555 (1:800, Invitrogen/ Life technologies). For binding experiments with CXCL16, cells were incubated with directly Cy3-labeled CXCL16 or lactalbumin (negative control with comparable molecular weight) at 4°C for 60 min in the dark, washed, fixed and nuclei counterstained with DAPI. Labeling was performed using monoreactive Cy3 NHS ester (GE Healthcare) following the manufacturer's instructions. Briefly, 2 µg protein was incubated with a four-fold excess of reactive dye in 0.2 M NaHCO₃, pH 8.4 (total reaction volume 90 µl). The reaction was stopped by addition of 1 µl 0.1 M Tris, pH 7.3. For binding experiments with CX3CL1, cells were incubated with biotinylated CX3CL1 or the control peptide (Fluorokine, R&D Systems) at 4°C for 60 min, washed, incubated with Alexa Fluor 488 streptavidin (Invitrogen/ Life technologies), washed again, fixed and nuclei were counterstained with DAPI. For flow cytometry analyses, cells were detached using 0,5 mM EDTA, stained with the CX3CL1 Fluorokine Assay following the manufacturer's advice. For CXCL16 experiments, recombinant CXCL16 (Pepro Tech) was biotinylated with the One-step antibody biotinylation kit (Miltenyi Biotech, Auburn, CA) and used (instead of biotinylated CX3CL1) with the Fluorokine kit components. Cells were analyzed using a FACSCanto System (BD Bioscience, Heidelberg, Germany). For electron microscopy, cells seeded on coated cover slips were pre-incubated in serum free DMEM (+0.5% fatty-acid free BSA) for 30 min at 37°C, then slowly cooled down (30 min room temperature, 15 min 8°C, 15 min 4°C). All following incubation and washing steps were performed at 4°C and with pre-chilled buffers and media. Cells were briefly washed (3x) with 145 mM NaCl, 5 mM KCl, 1.8 mM CaCl₂, 1 mM MgCl₂, 20 mM Hepes, pH 7.4, and incubated with the primary antibody (anti-CXCL16 or anti-CX3CL1, see above, 1:100) for 60 min, washed again and incubated with the secondary antibody (goat anti-rabbit with adsorbed 15 nm gold (Au) particles, diluted 1:40, British BioCell International, Cardiff, UK) for 60 min. Cover slips were washed again and incubated with 3 µg/ml either recombinant CXCL16 with adsorbed 5 nm gold particles or with biotinylated recombinant CX3CL1 (R&D Systems) and subsequent with streptavidin adsorbed to 5 nm gold particles (British BioCell International). Cells were fixed with 4% glutaraldehyde/ 0.5% paraformaldehyde, embedded in Araldite, sectioned and viewed on a Zeiss EM 900 electron microscope. Samples were only slightly contrasted (45 min 2% osmium tetroxide before embedding and 5 min uranyl acetate (saturated) after sectioning), as strong contrasting (e.g. with lead citrate) would not allow for clear detection of gold particles. Thus, contrast of electron micrographs was digitally

enhanced. Images shown are representative views of 3 (ICC) or 2 (FACS, EM) independent stimulations/ immunostainings.

## Cross-linking

After membrane isolation (Held-Feindt et al., 2010) proteins were dissolved in 0.2 M triethanol-amine-hydrochloride (pH 8.0), and 1.5 mg membrane-protein was incubated with 2 nM recombinant *s*-chemokine (PeproTech) over night at 4°C under slow shaking conditions. Subsequently 1% paraformaldehyde was added for cross-linking and incubated for 1 hr at room temperature. The reaction was stopped with addition of 1 M Tris (pH 8.3) and incubation for 15 min. Electrophoresis and Western blotting was performed as described above (rabbit anti-CXCL16 /anti-CX3CL1, 1:500; Pepro-Tech), representatives from 2 independent experiments, respectively, are shown.

## siRNA silencing

After cultivation of glioma cells (primary cultures and established cell lines) in DMEM plus 10% FCS in 6-well dishes (150,000 cells /well) for 24 hr, cells were transfected with siCXCL16 RNA or siCX3CL1 RNA (CXCL16 siRNA ID: s33808; CX3CL1 siRNA ID: s12630; both 50 pmol/well; Life technologies) dissolved in a mixture of Opti-MEM Medium and lipofectamine (Life technologies) for 6 hr. In parallel a transfection with silencer select negative control siRNA (Life technologies) was performed under same conditions. After transfection cell culture medium was changed and glioma cells were cultured for another 24 hr in DMEM plus 10% FCS. Then, cells were washed 20 min for three times with DMEM plus 0.5% FCS and afterwards stimulated for 15 min with recombinant CXCL16 or CX3CL1 (10 nM; PeproTech) dissolved in DMEM plus 0.5% FCS. Cells were lysed and applied for Western Blot experiments as described above, representative data from 2 independent stimulations of cell lines, and 2 different patients' derived primary cultures are shown.

## Migration, proliferation and anti-apoptosis assays

Migration was analyzed in wound healing assays (scratch assay, Hattermann et al., 2008). Briefly, 200,000 cells/well were seeded on 6-well dishes, grown to confluence, scratched with a pipet tip, washed and supplemented with stimuli, media with 10% fetal calf serum served as positive control. In each experiment, three scratch regions were photographed at 0 and 24 hr. Scratch areas were measured and differences between 24 and 0 hr were determined (yielding the settled area). Stimuli were normalized to non-stimulated controls. To measure proliferation, 5000 cells/well were seeded on 96 well plates and grown for 24 hr. Then media were changed to DMEM containing 1% BSA (plus respective stimuli or 10% fetal calf serum as positive control). After 24 hr incubation, proliferation was determined by the measurement of tetrazolium salt WST-1 cleavage (Roche, Mannheim, Germany) and normalized to non-stimulated control (3 individual wells for each stimulus). To investigate reduction of apoptosis, 300.000 cells were seeded in 25 $mm^2$ culture flasks and cultured for 2 days to reach confluency of 80%. Apoptosis was induced by addition of Temozolomide (400 µg/ml, Sigma-Aldrich) or 0.1 µg/ml Camptothecin applied in a stock solution in DMSO; the final solvent concentration of 2% (Temozolomide), 0.1% (Camptothecin) or 0.1 µM Staurosporine in cultures was also used in controls. After 48 hr, caspase-3/7 activity in glioma cells was measured with 40 µM Ac-DEVD-AMC (AMC, 7-amino-4-methylcoumarine; Bachem, Bubendorf, Switzerland) after lysis in 100 mM NaCl, 0.1% CHAPS, 10 mM dithiothreitol, 1 mM EDTA, 10% glycerol, 50 mM Hepes, pH 7.4. Alternatively, 18 hr after stimulation cleavage of Poly (ADP Ribose) Polymerase (PARP) was measured by Western blot (150,000 cells/25 $mm^2$ flask, grown for 30 hr and stimulated for 18 hr) or immunocytochemistry (30, 000 cells/cover slip, grown for 30 hr and stimulated for 18 hr) as described above using an antibody specifically detecting cleaved PARP (Asp124, 1:500 for WB, 1:100 for ICC; Cell Signaling). An antibody against GAPDH (1:500; Santa Cruz Biotechnology) served as loading control for Western blot, the immunocytochemistry signal obtained by fluorescence microscopy was measured and normalized to the nuclear area yielding OD/nucleus area.

## Statistical analysis

Values are given as means ± standard deviations (SD) of independent biological replicates, respectively. Diamonds shown in figures correspond to the data of an independent biological replicate, which means the experiment was performed with cells of a different subculture at a different time

point. Statistical significance was analyzed by a two-tailed Student's t-test. *p<0.05, **p<0.01, ***p<0.001.

## Acknowledgements

We thank Judith Becker, Martina Burmester, Sonja Dahle, Fereshteh Ebrahim, Marion Kölln, Jörg Krause, Miriam Lemmer and Kathrin Neblung-Masuhr for expert technical assistance and Clemens Franke for drawing figures. Prof. Dr. Udo Schumacher (Anatomy, University of Hamburg, Germany) generously supplied us with tumor cell lines. We thank Prof. Dr. Philippe Deterre (Laboratoire Immunité et Infection, INSERM UMR-S 945, UPMC Paris 6, Faculté de Médcine Pitié-Salpêtrière, Paris, France) for gift of recombinant CX3CR1-antagonist F1. This work was supported by the "Deutsche Forschungsgemeinschaft (ME 758/10-1, HE3400/5-1 and LU869/6-1)", by the popgen 2.0 network [(P2N; supported by a grant from the German Ministry for Education and Research (01EY1103)] and an intramural grant of the Medical Faculty of the University of Kiel.

## Additional information

### Funding

| Funder | Grant reference number | Author |
| --- | --- | --- |
| Deutsche Forschungsgemeinschaft | LU869/6-1 | Ralph Lucius |
| Deutsche Forschungsgemeinschaft | HE3400/5-1 | Janka Held-Feindt |
| Bundesministerium für Bildung und Forschung | 01EY1103 | Janka Held-Feindt |
| Deutsche Forschungsgemeinschaft | ME 758/10-1 | Rolf Mentlein |

The funders had no role in study design, data collection and interpretation, or the decision to submit the work for publication.

### Author contributions

KH, JHF, Conception and design, Acquisition of data, Analysis and interpretation of data, Drafting or revising the article; HG, Acquisition of data, Analysis and interpretation of data, Drafting or revising the article; SK, RL, Acquisition of data, Drafting or revising the article; AL, Conception and design, Drafting or revising the article, Contributed unpublished essential data or reagents; RM, Conception and design, Analysis and interpretation of data, Drafting or revising the article

### Ethics

Human subjects: All patient samples were obtained in accordance with the Helsinki Declaration of 1975 with approval of the ethics committee of the University of Kiel, Germany (file reference: D 442/11) and after written consent of donors.

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
