## [Decision Letter]

Thank you for submitting your work entitled "Transmembrane chemokines act as receptors in a novel mechanism termed inverse signaling" for peer review at *eLife*. Your submission has been favorably evaluated by Charles Sawyers (Senior editor) and three reviewers, one of whom is a member of our Board of Reviewing Editors.

The reviewers have discussed the reviews with one another and the Reviewing editor has drafted this decision to help you prepare a revised submission.

Summary:

Hattermann and colleagues describe results from a series of studies investigating the role of trans membrane chemokines CXCL1 and 16 to signal in a novel transmembrane mechanism. The authors provide data that shed chemokines can act in an autocrine and/or paracrine manner. They termed this mechanism "inverse" signaling. They put these results in the context of tumor biology and suggest it is a previously unrecognized of cell to cell communication. The signaling mechanism identified is novel, with a wealth of evidence of its functionality in vitro, in several cell line models. Cells negative for the classical receptors respond with the soluble chemokines in the presence of the membrane bound forms, non-responsive cells will respond to the soluble chemokines after transfecting the cells with the transmembrane forms, and inactivating the transmembrane forms by deleting their c-terminus, responsible for the intracellular domain diminish the signal transduction. Furthermore the authors showed the binding of the soluble forms to the transmembrane forms, and also other evidence for the novel signaling is provided.

Essential revisions:

Overall, the studies are well described and performed. Clearly, the results are novel and will likely be of interest. All three reviewers agreed on the interest in the studies and the potential significance of the findings. The biggest criticism is that no parallel in vivo data is provided. This is somewhat limiting, especially as the authors discuss the results in the context of tumor biology. Ideally this could be addressed with experimental evidence. At a minimum, this needs to be put in better context.

There were a few concerns raised about the in vitro studies that need to be addressed.

1) Why are the levels of the CXCL16 and CX3CL1 similar in all cells where they are expressed, like the signal is on/off- type?

2) Why would all cell types respond with equal robust readouts? No cell-specific responses were provided, and I feel this to a bit odd.

3) Figure 4 – In contrast to the stated conclusion, inhibition of CXCR7 appears to block the effects of the soluble chemokines (as there is not much more ERK phosphorylation compared with control when treated with inhibitor, compared with when the cells are treated with vehicle). I would advise repeating this assay to establish their conclusion with certainty.

This is a bit concerning because CXCR7 is a canonical receptor and its blockade should have no effect whatsoever since the whole premise of the paper is that canonical receptors are *not* required for inverse signaling. Thus it might be important to get them to sort this out.

---

## [Author Response]

*Essential revisions:*

Overall, the studies are well described and performed. Clearly, the results are novel and will likely be of interest. All three reviewers agreed on the interest in the studies and the potential significance of the findings. The biggest criticism is that no parallel in vivo data is provided. This is somewhat limiting, especially as the authors discuss the results in the context of tumor biology. Ideally this could be addressed with experimental evidence. At a minimum, this needs to be put in better context. There were a few concerns raised about the in vitro studies that need to be addressed. 1) Why are the levels of the CXCL16 and CX3CL1 similar in all cells where they are expressed, like the signal is on/off- type?

This question is indeed very interesting as this point might hint to a tissue/environment specific restriction of the “inverse signaling” and is therefore linked to the next reviewers’ comment. To address this question (expression of *tm-*chemokines = on/off?), we now show more ICC stainings for CXCL16 and CX3CL1 of selected cell lines in the bottom of Figure 1. These micrographs clearly indicate that expression levels are different in the respective (tumor) cell lines: while the brain tumor cell lines (glioma, neuroblastoma) show strong expression patterns, the breast cancer cell line MCF-7 shows only slight protein expression of the transmembrane chemokines. The LOX melanoma cell line does not express CXCL16 or CX3CL1 on protein level. Corresponding text passages have been added to the Results and the figure legend. Thus, protein expression corresponds to the mRNA expression levels obtained by qRT-PCR. However, these data are presented in a log-scale in Figure 1 (top). Thus, expression differences shown in the graph appear to be smaller than they are. To facilitate the readability of these data, we have added an explanation on ΔC_T_ values (a 3.33 alteration of ΔC_T_ value corresponds to a ten-fold altered expression) in this part of the Results section.

Together with the reviewers’ next comment on the cell specificity of the cellular responses this comment led us to the obvious question if the low but detectable expression level of MCF-7 cells affects the existence and dimension of the inverse signaling. Thus, we have performed some additional experiments addressing the activation of the MAP kinase pathway and the rescue from apoptosis also with MCF-7 cells, and added the respective results to the Supplementary material (Figure 3—figure supplement 2). Indeed, the *tm*-chemokine low expressing cell line MCF-7 shows only slight activation of MAP kinase signaling upon stimulation with *s*-chemokines, and the rescue from apoptosis is only mild, and hardly robust.

On this note, the data hint to a cell type specific expression strength rather than emphasizing an on/off signal. This is also supported by observations by us and others that both *tm*-chemokines are induced by pro-inflammatory cytokines, and therefore can be regulated by environmental conditions. A corresponding remark citing some examples in the literature has been added to the Discussion.

*2) Why would all cell types respond with equal robust readouts? No cell-specific responses were provided, and I feel this to a bit odd.*

This point is – as already explained above – related to the previous point as a specific response may on the one hand depend on the cell type or on the other hand on the respective expression level. Concerning different cell lines, we see different activation kinetics of the MAP kinase in different glioma cell lines upon stimulation with the *s*-chemokines (compare Figure 2). To, however, more properly address the raised concern, we performed stimulation experiments with a breast cancer cell line, MCF-7, expressing low levels of *tm*-chemokines. As mentioned above, this cell line showed also activation of the MAP kinase pathway and a rescue from apoptosis. However, these effects were clearly smaller and less robust in comparison to the *tm*-chemokine high expressing glioma cell lines. These data were added as supplementary material to Figure 3 (Figure 3—figure supplement 2), and a note was added to the Results.

Apart from this, in a separate manuscript, we show responses and effects of “inverse signaling” in meningioma cells that closely mirror the cellular effects of glioma cells as described in this manuscript. Briefly, primary human meningioma cells (meningioma: benign brain tumor descending from the meninges) showed activation of the MAP kinase pathway, enhanced proliferation and a reduction of caspase activity upon stimulation with CXCL16.

Taken together, our data emphasize an activation of the MAP kinase pathway and anti-apoptotic/proliferative effects of the “inverse signaling” of *tm*-chemokines that depends on the expression level of the *tm*-chemokines rather than the respective (tumor) cell type.

*3) Figure 4 – In contrast to the stated conclusion, inhibition of CXCR7 appears to block the effects of the soluble chemokines (as there is not much more ERK phosphorylation compared with control when treated with inhibitor, compared with when the cells are treated with vehicle). I would advise repeating this assay to establish their conclusion with certainty.*

*This is a bit concerning because CXCR7 is a canonical receptor and its blockade should have no effect whatsoever since the whole premise of the paper is that canonical receptors are not required for inverse signaling. Thus it might be important to get them to sort this out.*

To more substantiate this, we replaced the western blot in Figure 4 with a more convincing experiment, and added replications of this experiment to the supplementary figures (Figure 4—figure supplement 1). Additionally, to explain the phenomenon of non-canonical chemokine receptor signaling for a broader readership we have addressed this point in the Discussion.